# HESS Opinions. Participatory Digital Earth Twin Hydrology systems (DARTHs) for everyone: a blueprint for hydrologists

Riccardo Rigon[1], Giuseppe Formetta[2], Marialaura Bancheri[3], Niccolò Tubini[2], Concetta D'Amato[1], Olaf David[4], and Christian Massari[5]

[1]Center Agriculture Food Environment - C3A, University of Trento, Trento, Italy
[2]Department of Civil, Environmental and Mechanical Engineering - DICAM, Trento, Italy
[3]Institute for Mediterranean Agricultural and Forestry Systems (ISAFOM), National Research Council (CNR),Italy
[4]Department of Civil and Environmental Engineering, Colorado State University, Fort Collins, CO, USA
[5]National Research Council, Research Institute for Geo-Hydrological Protection, Perugia, Italy

**Correspondence:** Riccardo Rigon (riccardo.rigon@unitn.it)

**Abstract.** The Digital Earth (DE) metaphor is very useful for both end users and for hydrological modellers (i.e., the coders). In this opinion paper, we analyse different categories of models, with the view of making them part of a Digital eARth Twin Hydrology system (called DARTH). We stress the idea that DARTHs are not models, rather they are an appropriate infrastructure that hosts (certain types of) models and provides some basic services for connecting to input data. We also argue that a
5 modelling-by-components strategy is the right one for accomplishing the requirements of the DE. Five technological steps are envisioned to move from the current state of the art of modelling. In the first step, models are be decomposed into interacting modules with, for instance, the agnostic parts deal with inputs and outputs, separated from the model specific ones that contain the algorithms. In the other three steps, appropriate software layers are added to gain transparent model execution in the cloud, independently of the hardware and the operating system of computer, without human intervention. There is a fifth step that
would allow models to be selected as if they were interchangeable with others without giving deceptive answers. This step includes: use of hypothesis testing, inclusion of error of estimates, adoption of literate programming, and guidelines to obtain informative clean code.

The urgency for DARTHs to be Open Source is supported here in light of the Open Science movement and its ideas. Therefore, it is argued that DARTHs must promote a new participatory way of doing hydrological science, where researchers can
contribute cooperatively to characterize and control model outcomes in various territories. Finally, three enabling technologies are also discussed in the context of DARTHs, i.e., Earth Observations (EO), High Performance Computing (HPC) and Machine Learning (ML) and how they can be integrated in the overall system to both boost the research activity of scientists and generate knowledge.

## 1 Introduction

The Digital Earth (DE) concept was first developed by the US Vice President Al Gore in a speech for the opening of the California Science Center in 1998. In Al Gore's vision, the DE is meant to be "a multi-resolution, three-dimensional representation of the planet into which we can embed vast quantities of geo-referenced data". Although in 1998 the technologies available were not at all adequate to pursue this vision, Al Gore's speech aroused great interest in the scientific community and in 1999 the first International Symposium on Digital Earth was held in Beijing China. The outcomes of this symposium were summarized in the 1999 Beijing Declaration on Digital Earth. In 2006, the International Society for Digital Earth (ISDE) was formally established, and in 2008 it gave itself an international peer-reviewed academic journal, when the International Journal of Digital Earth was founded (IJDE). Since then, another milestone document, the 2019 Florence Declaration on Digital Earth, was approved and position papers published (Goodchild et al., 2012; Craglia et al., 2012; Guo et al., 2019).

Over the years the original vision has continually evolved, following the progress of technologies and of the global community of interest. Nowadays, the DE is considered a global strategic contributor to scientific and technological developments, and it could play a strategic and sustainable role to face issues characterizing the Anthropocene epoch. Thanks to the advancement of technologies, it is possible to talk about Digital Twin Earth models (DT). They want to accurately reproduce the state of evolution of a generic physical entity through high-fidelity computer models, with the aim of understanding, simulating systems behaviors, and evaluating them under changing boundary and initial conditions. As such, the DT dates back to NASA's Apollo program (Semeraro et al., 2021) and, before, they were developed in sectors where the processes being modelled are well understood, such as industry and manufacturing (Graessler and Pöhler, 2017).

In modelling natural processes, DE models use the same concepts (e.g., physical principles, equations) and data as Hydrology. It is worth noting that as early as 1986 Eagleson recognized the need to develop hydrological models to produce hydrological prognoses at the global scale (Eagleson, 1986). These global-scale hydrological models (Beck et al., 2017; Emerton et al., 2016; Stacke and Hagemann, 2021; Döll et al., 2003), lack the "organizing vision" of DE, as they are not an integral part of other advanced technologies, such as earth observation, geo-information system, virtual reality, and sensor webs, for example.

Currently, the convergent forces of the Space Agencies and the products of their satellites have given substance to the possibility of really getting hyper-resolution models of Earth (Wood et al., 2011). These efforts have also triggered the interest of the computer science community, with its high-performance, distributed computing infrastructures and all the related technologies. However, these same efforts often bring to light an approach where research and its related data are created and shaped by big institutional players, even private entities, and not by the large majority of researchers. At the same time, small research groups and researchers simply do applications using already deployed tools, unless they are directly involved in the development. As such, this top-down approach could limit creativity and the possibility of a vast community to contribute to the advancement of science and innovation (Oleson et al., 2013; Best et al., 2011).

The EU's recent Digital Twin Earth campaign (e.g. https://digital-strategy.ec.europa.eu/en/policies/deearth) runs the same risk: it is time therefore to restore a bottom-up approach in which the creativity of individuals and small groups can be harmonized within the big view, with the least number of institutionalized organizations possible.

This paper looks at this goal from the perspective of those who deploy "community" hydrological models, meaning models developed by a community of researchers that freely gathers and discusses ideas about hydrological and Earth system science, produces model parts and commits them to common, decentralized repositories. Their contribution usually encompasses theoretical achievements, implementation design, science verification from data, and deployment for applications, which are seen as the natural outcome and source of the most fundamental research of hydrologic processes.

This opinion paper aims, therefore, to:

- provide an up-to-date analysis of the current research in designing and developing DE components, with a particular focus on hydrological modelling;

- identify possible directions and hints, based on more than a decade of experience in deploying hydrological models (Rigon et al., 2006; Endrizzi et al., 2014; Formetta et al., 2014; Tubini and Rigon, 2021) and contributing to GIS developments (JGrass, uDig, gvSIG);

- debate and identify potential ways forward to answer the following questions: is it possible to have more than one Digital eARTh Hydrological (DARTH) model while avoiding fragmentation of efforts? How can data become available to allow the vision to be incrementally realized? Should the models be designed differently? Which informatics is suitable? How should high performance computing (HPC) be envisioned and developed? What efforts should be made in modelling and data representation? How is the scientific reliability assessed ? What role can Earth observations (EO) play? Is Machine Learning (ML) the solution?

The paper is organized in four sections, with many subsections, and an appendix, and it has a few accompanying manuscripts. Section 1.1 discusses the idea that a DARTH is not a single piece of software but an ecosystem of contributions; section 1.2 briefly discusses the questions that regard data availability and data flow. They represent an introduction to the DARTHs topic.

Section 2 deals with design and implementation requirements. Its subsections first introduce and then explore the suitable modelling architecture for the purpose of buildings DARTHs components. Reusability and the possibility of changes to scientific paradigms, with as few restrictions as possible, are presented as key DARTHs concepts. Since good programming practices are of fundamental importance for open science, this section also contains a discussion on clean coding, literate programming and, as a consequence, literate computing. Subsection 2.7 raises the topic of the reliability of DARTHs.

DARTHs are themselves among the "enabling technologies", i.e., innovations that can drive radical change in the capabilities of scientists to do new hydrology. Sections 3.1 to 3.3 deal with three other of these technologies that are pervasive in contemporary sciences and can be seen as protagonists in the near-future developments: HPC, Earth Observations (EO) and the use of Machine Learning (ML) techniques, providing for each a short review of applications, issues and perspectives. Section 4 discusses what could be the governance and organization of communities of DARTHs developers. Conclusions follow. Finally,

the supplemental material contains: a glossary of terms (where we define and explain acronyms and jargons used in the paper), with a specialized glossary regarding a classification of models from the point of view of DARTHs; a DARTHs cheatsheet that summarizes the contents of this paper; the results of a survey done among model and model infrastructures developers that tries to capture the present state of the art. Because the paper makes use of many acronyms, all of them are reported in Table A1.

## 1.1 From models to DARTHs

Since there are no real DEs in hydrology to date, our analysis refers to the most used hydrological models as a starting point. Their history (Beven, 2012) shows that there is a fragmentation of models and "legacy more than adequacy" is the rule for researchers when choosing an application (Addor and Melsen, 2019). Models like SWAT (Arnold et al., 2012; Neitsch et al., 2011), HEC-HMS (Chu Xuefeng and Steinman Alan, 2009), SWMM (Gironás et al., 2010), and the Topmodel family (Peters et al., 2003) or the good old reservoir models (Knoben et al., 2019) have the lion's share among users. These models usually provide a good balance between usability and reliability (in the sense that they produce plausible patterns and have no or few running issues). The current state of the hydrological science allows us to say that all these models work, at least for the purposes they were requested, but they are normally closed to easy modification and tend to lag behind the state of the art of hydrological studies. The state of the art, on the other hand, often makes use of artisanal products, restricted by badly engineered codes, in the hands of a few researchers and it is not thought for the general reuse that the DE paradigm would require. Many modular frameworks have been recently built with the aim to filling this gap, like SUMMA (Clark et al., 2011a), SuperPyflex (Dal Molin et al., 2021), GEOframe (Formetta et al., 2014), RAVEN (Craig et al., 2020). However, from the points of view of both researchers and users, a further quantum leap must be made in model infrastructure design and model implementations, to cope with the DARTHs requirements that are addressed in the paper.

Broadly speaking, four mathematical tools dominate in hydrological modelling research (Kampf and Burges, 2007): the process based models (PB) (Paniconi and Putti, 2015; Fatichi et al., 2016a), the reservoir type models (HDS, as Hydrological dynamical systems) (Todini, 2007; Bancheri et al., 2019b), the classical statistical models (McCuen, 2016), and the current algorithmic-statistical models that make use of one form of machine learning (Shen et al., 2018; Levia et al., 2020). To these we could add a further type of model which is a tight black-box between Earth-Observations and lumped models and/or machine learning models usually referred as EO products (Martens et al., 2016). Many references already discuss the taxonomy of models (e.g., Kampf and Burges (2007)), and the strengths and weaknesses of each of these approaches (Hrachowitz and Clark (2017)), and their application at various scales of application, and we do not want to add further to those analyses. We just adopt the pragmatic view that they exist, that, notwithstanding their uncertain informatics quality (except maybe for the ML tools that are based on the use of large frameworks), they are used creatively, and they still continue to produce insights that solve hydrological issues.

All of these models rely on parameter calibration (Duan et al., 2003) or on some type of "learning" to make their predictions realistic (Tsai et al., 2021). At present, it is impossible to disentangle the complexity of the model variety and make it simpler. The matter of Hydrology and Earth system sciences is complex (in the sense of the complexity sciences) and complicated

(affected by huge variability and heterogeneity) and whatever it takes to get a clue should be welcomed. This statement, however, implies that we cannot have a DARTH in which just one "solution" is adopted, rather we need a DARTH where many, even competing, paradigms (like PB modelling, ML tools and traditional lumped models) can be tested and compared, and eventually put to work together to assess the uncertainty in forecasting. This, in turn, has consequences on the code architectures, infrastructures and the informatics that have to be deployed. Some naive ideas that ML plus EO, for instance, could do it all, like some Google Earth-like applications seem to envision, is just wishful thinking that clashes with the current view we have of the discipline. Instead, the idea that different models (or model structures) have to be used and subjected to hypothesis testing procedures has gained ground (Clark et al., 2011b; Blöschl, 2017; Beven, 2019), even in ML (Shen et al., 2018).

Recently (Gharari et al. (2021)), in fact, it has become clear that ML and Deep Learning (DL) techniques can be interpreted and explained, thus they can be used as a tool for understanding (Arrieta et al., 2020) or model parameter learning (Tsai et al., 2021), instead of primarily for predictive purposes. In any case, ML growth has been mostly driven by a large variety of problems, for instance, computer vision applications, and speech and natural language processing in a way that has to be harmonized with the practices of more traditional ways of conceiving hydrological models.

If the model paradigms cannot be compressed in favour of one choice, then a suitable DARTH engine should allow for the implementation of various ones, and more than one competing DARTHs engine should be made available in the community because the implemented technologies also come with their own legacies.

Just as life on Earth is built upon the four nucleotide bricks and presents an immense variety, it is clearly possible that DARTHs can share common protocols, standards and features and evolve, picking up the best without a continuous recreation of the whole infrastructure from scratch. That is to say, promoting diversity should be accompanied by the sharing of standards for the parts. The way to do it has been traced, for instance, in Knoben et al. (2021) where it is argued that the whole models informatics can be separated into model-agnostic parts, which potentially can be shared, and model specific parts, which could be differentiated among the various developers or research groups.

It should be stressed, however, that according to their own definition DARTHs are not simply hydrological models. Models made by scientists are usually developed to test their theoretical works or to provide evidence for research publication; as such, they do not have software quality as their primary goal. DARTHs have models at their scientific core but, according to the vision we promote, they need to provide other services too:

– be available on demand, working seamlessly on the cloud as web services

– have automatic ways of retrieving and providing data.

– and be interoperable with other models;

In order to achieve these goals, the modelling needs to be supported by appropriate layers of software that orchestrate the whole functioning. Besides,

– they need to be implemented robustly (see the glossary) and properly designed to be fault-tolerant.

In this paper we also support the idea that DARTHs have to serve science in its doing and evolving, harmonizing the work of researchers that develop and use them.

A good practice in object oriented programming is the *separation of concerns* (Gamma et al., 1995), which states that any class should have possibly only one responsibility. This approach seems reasonable in this context too, and in the following we will try to break down the DARTHs into their main compositional parts.

## 1.2  A necessary first step: making data and formats open

DARTHs, obviously, are useless if there are no data available to run them; therefore, we discuss briefly some of the data aspects relevant to envisioning their design. There is a frequent lack of institutional and political will to publish environmental data, sometimes for reasons of national security, other times to prevent misuse or for confidentiality. This state of affairs creates many obstacles and open spatial data infrastructures (open SDIs) are still not common (Nedovic-Budic et al., 2011; Lehmann et al., 2014). This contrasts with the literature that, for some years now, has forecast a data-rich era for hydrology, coining neologisms like datafication (Mayer-Schönberger and Cukier, 2013) with reference to the upcoming ubiquitous presence of the Internet of Things (IoT) even in natural contexts. However, in the U.S. environmental data are public and supported by various initiatives; in other countries, i.e., in Austria (e.g., https://ehyd.gv.at), France (e.g., https://www.hydro.eaufrance.fr/), Germany (e.g., www.ded.de), U.K. (e.g., https://archive.ceda.ac.uk) there are projects that aim to bring the largest possible number of data to be open to users but data availability sometimes requires the payment of a fee. Since the survey by Viglione et al. (2010), there have been various initiatives to fill the gaps by various institutions, such as Global Runoff Data Center (GRDC, http:www.bafg.de/GRDC), the Global Water Monitoring System (GWMS: http://www.gemstat.org), the Global Earth Observation System of Systems (GEOSS:https://earthobservations.org/geoss.php), the WMO Hydrological Observing system (WHOS), and, more recently, the Destination Earth architecture. It is also worth mentioning the bottom-up effort started with the CAMELS datasets, so named from the acronym from Catchment Attributes and MEteorology for Large-sample Studies (Addor et al., 2017). Meanwhile, the Open Geospatial Consortium has seeded several projects (https://www.ogc.org/node/1535) to establish standards for the delivery and deployment of data related to Hydrology. A review of initiatives, topics and issues can be seen in Lehmann et al. (2014) and visionary perspectives are presented in Nativi et al. (2021), to which we cannot add more. It is clear, however, that the general situation is still largely placed in between "fragmentation and wholeness" (Ballatore, 2014); this can only be resolved with general agreements and consensus about the usefulness of sharing data and time, wiyhout forgetting the question of providing the appropriate ground based data for the needs of developing countries .

To complicate the matter, the new developments in satellite remote sensing have been producing massive amounts of data at unprecedented spatial and temporal resolutions. According to statistical data from the Committee on Earth Observation Satellites (CEOS), over 500 EO satellites have been launched in the last half-century, and more than 150 new satellites will be in orbit in the near future (CEOS, 2019). For instance, data from missions like Sentinel of the new European Copernicus Earth observation programme (https://sentinels.copernicus.eu) have already exceeded petabytes of volume. More and more data are expected to come: the American, Japanese, Indian, Chinese, and European space agencies have already planned new missions for observing the planet. Notable examples are the NASA Cyclone Global Navigation Satellite System (CYGNSS)

(Chew and Small, 2018) and SWOT missions (https://swot.jpl.nasa.gov/mission/overview/), the US-Indian NASA-ISRO SAR (NASA-ISRO, 2018), the Radar Observing System for Europe—L-band (ROSE-L) (Davidson et al., 2019) SAR missions and LSTM missions, and the Thermal InfraRed Imaging Satellite for High-resolution Natural resource Assessment (TRISHNA) (Lagouarde et al., 2019), in addition to a constellation of low-cost commercial CubeSats satellites (McCabe et al., 2017a).

So the question of how these data can be easily accessed and used by DARTHs is not a trivial one. Interesting solutions to facilitate this already exist, such as the Google Earth Engine (https://earthengine.google.com) and the new born European Open EO (https://openeo.cloud). These earth observation data centres have revolutionized the way users interact with EO and offer integrated solutions to access large volumes of a variety of EO data with relatively high speed. With this new paradigm of "bringing the users to the data", users no longer need to download and store large volumes of EO data and they do not have

the problem of dealing with the different formats and grids made available by the range of data providers. DARTHs should therefore provide an easy integration with these new platforms to facilitate the use of the wealth of data made available by the various Earth observation programmes.

    The state of art of the matter, for what can be important for DARTHs, is that data layers have to be somewhat loosely coupled to models through a brokering approach, where some intermediate tool takes care of discovering, gathering and delivering data

according to what is requested. This approach has been successfully tested in the recent past (Nativi and Bigagli, 2009; Nativi et al., 2013) and allows the binding of heterogeneous resources from different providers. It is, for instance, at the core of the Destination Earth infrastructure and data space https://digital-strategy.ec.europa.eu/en/policies/destination-earth.

    A recent conference contribution (Boldrini et al., 2020) cites a few examples of brokers: WHOSWHOS; CUAHSI HydroDesktop (through CUAHSI WaterOneFlow); National Water Institute of Argentina (INA) node.js WaterML client (through

CUAHSI WaterOneFlow); DAB JS API (through DAB REST API); USGS GWIS JS API plotting library (through RDB service); R scripts (through R WaterML library); C# applications (through CUAHSI WaterOneFlow); and UCAR jOAI (through OAI-PMH/WIGOS metadata). At present, both in research and in applications, the action of matching data and models is done off-line by the researchers, but the DARTH vision would require that these data be automatically ingested and processed. Therefore, DARTHs should take care of these aspects by design and be able to abstract data from the algorithms, as also

explained in (Knoben et al., 2021).

    Regarding the data sharing/use, and the fact that chunks of DARTHs could be distributed for inspection and modification by third parties, two strategies could be used. In the ideal situation where there is a universal broker that, as a provided service, collects required data from the cloud in real time, DARTHs should include a connector to this broker so that all modelling could be performed smoothly (the cloud strategy). The opposite approach is that data is acquired once and for all and locally stored

(the local strategy). The local strategy is clearly unfeasible, due to the burden of data, while the cloud strategy is clearly still immature, even if ML has recently introduced new paradigms in data treatment such as, for instance, the distributed platforms Hadoop or Apache Spark (Nguyen et al., 2019) to facilitate this type of operations.

    At a lower level of implementation, the issue of data treatment is managed through the standardization of formats. Standards have been developed for various scopes, including grib (Dey and Others, 2007), NetCDF (Rew and Davis, 1990), HDF5 (Folk

et al., 2011) and database formats. Other, general purpose standards, like Apache Arrow (https://arrow.apache.org/) for tabular

| Roles / Users | Hard Coders | Soft Coders | Linkers | Runners | Player | Viewers | Providers |
|---|---|---|---|---|---|---|---|
| Prime | | | | | | ● | ● |
| Other End Users | | | | ● | ● | ● | ● |
| Technical | | ● | ● | ● | | | |
| Researchers | ● | ● | ● | ● | ● | | |

**Figure 1.** After Rizzoli et al. (2006) - Models serve diverse scopes and uses. Therefore they are a fit-for-purposes task.

data, can be an interesting choice and come with various interfaces to the most common programming languages, thus allowing for easier compatibility with existing models.

We are not claiming here that some formats should be preferable to others. However, some format, and some database architecture, should be chosen as a starting point along with a set of tools to transport data from the one format to others. The data format itself should be self-explicative and not require additional information to be understood. A long experience in this direction is available from UNIDATA (https://www.unidata.ucar.edu/), from meteorology WMO, from ESA and NASA. For any DARTH builder, therefore, it should not be difficult to agree on some conventional formats to start with coherently.

## 2 Design and implementations requirements

In this section we delineate which are the main design requirements for DARTHs. The starting point is observing the variety of users and use cases in order to have a comprehensive view of the matter.

### 2.1 Starting from the people: glimpses of information about DARTH core architecture

As Rizzoli et al. (2006) realized, there are different types of users and different types of scopes when modelling hydrology. They are summarized in Figure 1.

Hard coders (core coders) are those who actually design and write code. Soft coders are those who just modify existing codes and, if possible, develop plug-ins using public Application Programming Interfaces (API). Linkers are power users who assemble existing codes through scripting languages or procedures. This role is of particular importance in ML where very

often earth scientists build their models on top of ML engines, provided by big companies or institutions (Nguyen et al., 2019). Runners (users) are those who execute existing codes. They just modify inputs, outputs and parameters, and they create and define scenarios. Players run scenarios and make analyses. Viewers view the players' results, having a low level of interaction with the framework, and maybe use them to instantiate policies. Providers provide inputs and data to all other user roles (and what was said in the previous section applies to them). The users' names are self-explanatory. In this paper, we concentrate on the specific roles of coders, linkers and runners, and to technical and research users. We also have to mention that, even if hard coders seem the most basic of developers, they can be classified as "professional end user developers" (Kelly and Sanders, 2008). Scientists and engineers developing software have backgrounds in the theoretical models implemented in software, but they do not usually have a strong back-ground and formal training in computer science or software engineering (while vice-versa, computer scientists are not usually domain experts). The coders we are referring to here are not usually computer scientists, but domain experts, i.e. hydro-bio-geo experts, with limited computer science backgrounds.

To serve the needs of all of those roles and users, besides the scopes inherent to the DARTH vision itself, DARTHs need to be supported by an appropriately modular software infrastructure able to manage the different needs that each user and role has. In addition, in thinking of coders/researchers, a DARTH infrastructure should be able to accept different modelling styles and programming paradigm changes,

and DARTHs have to facilitate good programming and testing practices, bridging the knowledge gap of scientist/hydrologists in computer science. Having readable code is something that cannot be renounced (Riquelme and Gjorgjieva, 2021). It is a key aspect for a successful long-term development of the code: it improves maintainability and reusability (David et al., 2013), it saves time in the future, and eases the development and growth of the research community (Riquelme and Gjorgjieva, 2021). This topic is further discussed in section 2.4.

Since we are aiming to gather a community of developers to build and evolve a DARTH, the code must work seamlessly on the major operating systems, whether MS Windows, Mac OS, Linux or others, and this excludes efforts limited to one platform. Besides being programming language agnostic, the foreseen DARTHs have to be operating system agnostic. It seems a secondary limitation but, in the present state of the matter, it is certainly not, even if the DE paradigm envisions a type of infrastructure operating over the web that encompasses the characteristics of single machines.

Other consequences follow if we look at the current hydrological models from the coders' position: they are codes written in various programming languages, these days primarily in FORTRAN, C/C++, Python, R, Java, C#, Julia, and Matlab. The variety of programming languages used brings a legacy that cannot be avoided with a blip, excluding the great part of programming researchers that has been at the core of the evolution of the field. DARTHs need, therefore, to be based on a platform which, while not being language agnostic per se as it is built in a specific language, is able to link libraries in all the languages, or libraries themselves have to evolve to become (web) services, as often happens in ML. Only in this way, will it be able to approach the mass of researchers and make treasure of the efforts already made. This would bridge the past and future of hydrological and environmental modelling and make revolutionary changes possible in an incremental way. DARTHs should not subtract, just add and evolve.

There are at least two other matters about the model design requirements of DARTHs that have to be accomplished. The first is related to the specification of the data a model actually uses. We cannot identify the external behavior of a model by saying, for instance, that it needs "precipitation" but more precision is necessary about the units, and the spatial and temporal dimension. To continue with the given example, we need to specify the form of the precipitation (e.g., liquid water precipitation), the spatial aggregation (e.g., catchment average), the temporal resolution (e.g., hourly), and the units (e.g., mm). In fact, the problem was raised quite a long time ago when first there was the need to give metadata, explaining what the values in each variable represent, and which are their spatial and temporal properties. The CF convention (https://cfconventions.org/) is one of the attempts to clarify these aspects, followed by other initiatives such as the Basic Model Interface (Jiang et al., 2017; Peckham et al., 2013a), which contains a "grammar" for creating new names and is more specialized than CF in the hydrological sciences. The success of a DARTH depends on these specifications in which the freedom of choice is sacrificed to the adoption of some standard that makes the models potentially interoperable and discoverable in the web by their input and output requirements. Assuming that one such convention would be adopted by all models, a user would know which models they can connect to obtain the dynamics of a process they want to estimate or predict. A black box characterization of models can be done by accurately specifying their inputs and outputs (i/o).

The second is that researchers and the concerned users do not care just about the models i/o but also of their realism, reliability, (replicability-)reproducibility and robustness, characteristics whose meaning is discussed thoroughly in the glossary. This information, which, in part, can be tested a *posteriori* by use and comparison with real world observations, can also be gained by inspecting the models' codes with their implementation details.

To sum up, the DARTH architecture must account for various requirements:

– to be programming language agnostic;

– to work seamlessly under the various operating systems;

– to use standard, self-explanatory data formats;

– to use standard names for the quantities they treat in input and output.

## 2.2 Too complex systems do not serve the cause

Another characteristic deemed important for DARTHs infrastructure is that they should not be invasive of programming habits (i.e. not forcing the programming to adopt constructs that computer scientists like but scientists and engineers cannot manage). The available environmental modelling frameworks, which are infrastructures to all effects, can be classified into two broad categories: heavyweight frameworks and lightweight frameworks (Lloyd et al., 2011). The former is characterized by large and unwieldy API that require a considerable effort from developers (scientists or soft coders) to become familiar with before writing new code. Moreover, such an effort somehow creates a strict legacy within the infrastructure and this has limited the diffusion of these systems in the recent past (David et al., 2014).

Conversely, David et al. (2013) show how lightweight frameworks have many functions for the developer due to the techniques used to reduce the overall size of the API and the developer's dependence on it. A lightweight Environmental Modeling Framework (EMF) fits easily with existing models as there is no interference with complex APIs. This is very useful for environmental modellers as it allows the use of existing modelling code and libraries, integrating them into a larger framework. In a lightweight framework, model components can work and continue to evolve outside the framework, and so adopting and using a lightweight framework is easy. This type of infrastructure is the most suited to DARTH development as each research group is not strictly bound to a specific infrastructure and it becomes easy to include new modelling solutions in existing models.

## 2.3 Participatory needs

If replicability of results is guaranteed by the availability of models, data and knowledge of the simulation setups, the clearest level of knowledge can be obtained when the codes are open source and can be analyzed. Being open source is not a mandatory requirement but certainly helps the structuring of open science (Hall et al., 2021) and is highly desirable for the openness of science and code inspection by third parties. In fact, there is no such thing as "black-box science" (Stodden et al., 2013) and the peer review process alone is largely ineffective at "sniffing out" poor validation and testing of model contents (Post and Votta, 2005).

We are committed to open science in this paper and therefore, in the following, we take for granted that the codes are open source, provided in an open repository, with an open license and with the building tools (i.e., the tools to compile the source codes) as well.

If we then consider the point of view of the runners, we are interested in the reliability of the contents and that runners can give their findings back to the system and enrich the knowledge base.

A new paradigm can be formulated as "participatory hydrological modelling", indicating that it can be used by many researchers who can seamlessly cooperate to produce added value for the DARTHs. Not only do we aim to have a multitude of users, there should be a multitude of developers to exploit science bringing bottom-up contributions. These requisites have some consequences in the functional design of a model. Take for instance the case of the Object Modelling System (David et al., 2013): besides a fully configurable setup, it provides a standard organization of the files used for modelling, as presented in Figure 2. A runner can be provided with all they need (input data, parameters values, input and output file names, modeling solutions structure definition, etc.) to run their simulation (in the dist folder), in this case locally on their own computer, including the source code, the size of which is quite irrelevant with respect to the data (in "data" and "output" folders). The already prepared simulation is described and governed by ".sim" files in the "simulation" folder that contains the workflow of what is going to be executed with all the required information. The runner can then add new simulations, with changed parameters, modified spatial partitions and input data, and then distribute all of their improvements and results back to the community with which they are interacting. This procedure, yet not automatized, has been already widely used by the Authors in their projects where collaborative efforts were necessary. The idea can be improved, no doubt, and the organization provided is very simple, based on a file system organization; much more sophisticated architectures, using databases and web-services, could be deployed to obtain the same functionalities with automatic procedures, similar to those used in the version control system

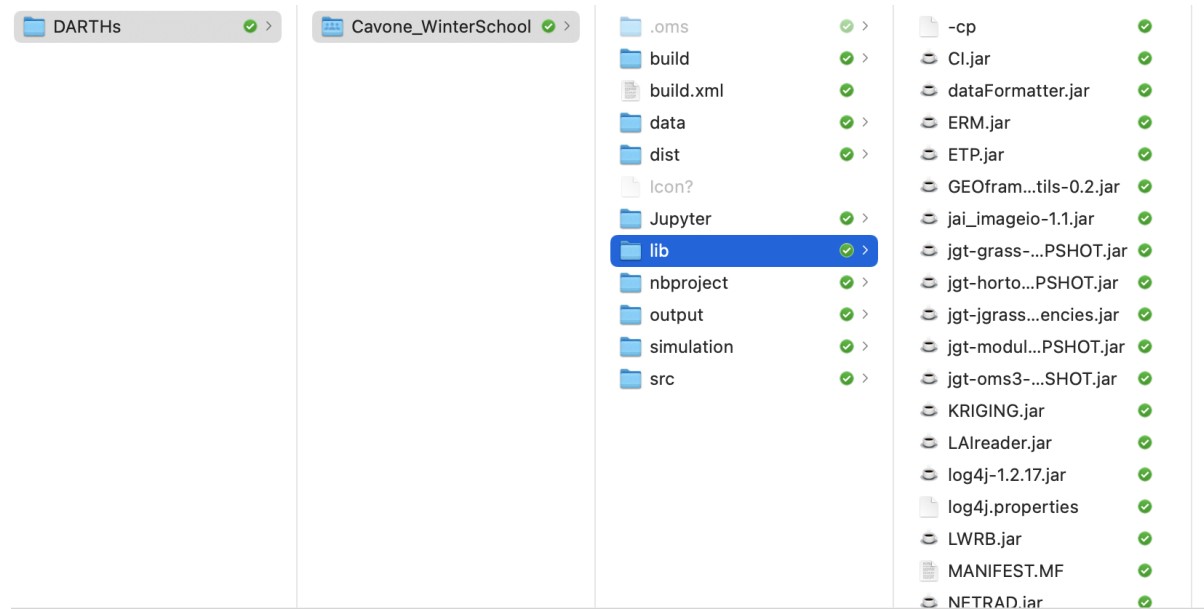

**Figure 2.** Standard organization of the OMS3 files with the libraries used highlighted

for codes. Then, if we assign a unique identifier to any catchment worldwide, the procedure can even be decentralized and certified by using, for instance, block-chain-like technologies (Serafin et al., 2021). But, simple as it is, it is a proof of concept that such a strategy of participatory modelling is possible.

### 2.4 Four, plus one, steps towards DARTHs

As described in the previous sections DARTHs can then be envisioned as a distributed ecosystem of softwares that serve multiple roles and users and where models and data can move around the web. To get some order within the models' design, computer scientists distinguish between various levels of interoperability, according to the openness, digital portability, and client-interaction style that characterize a given tool, i.e., a data/process driven analytical model provided as a digital software or service. Nativi et al. (2021) distinguish three levels, but we identify five with the following definitions (which are presented 345 with more detail in the glossary):

– Model-as-an-Application (MaaA): this is the case for most of the existing classical hydrological models (Rizzoli et al., 2006). Traditionally engineered models tend to be "applications", meaning that they bundle together all the features that are required to have an all round modelling experience, but these are exactly the opposite of what is needed by DARTHs, where everything must be provided as a service and loosely tied. Knoben et al. (2021) give a further clear description of 350 MaaA. Most current models fall into this category.

– Model-as-a-Tool (MaaT): it is an evolution of MaaA that allows interaction with the model. Actually the Runner interacts with a software tool, i.e. an interface, developed to utilize the model and not with the model itself, or a service API. The

implementation of the model runs on a specific server (or locally) and it is not possible to move the model and make it run on a different machine. Just a few of the recent modularized systems fall into this category, as can be deduced from the survey we present in the supplemental material.

- Model-as-a-Service (MaaS): as for the previous case, a given implementation of the model runs on a specific server, but this time APIs are exposed to interact with the model. Jupyter (Loizides and Schmidt, 2016) itself can be seen as an infrastructure that promotes such an approach, even if some functionalities are missing. Also the recent eWAterCycle (Hut et al., 2021) collector of models can function this way. OMS3/CSIP (David et al., 2013) and Craig et al. (2020) are other examples that almost fully implement this type of infrastructure.

- Model-as-a-Resource (MaaR): the interoperability level follows the same patterns used for any other shared digital resource, like a dataset. This time, the analytical model itself (and not a given implementation of it) is accessed through a resource-oriented interface, i.e., API. Besides, there is a software infrastructure layer, uploadable on heterogeneous hardware, that manages (not directly by the user) a set of compliant models. This software layer allows one to move models and make them run on the machine that best performs for a specific use case among a large pool of computing facilities without the need of user intervention (at the model level).

MaaA, MaaT, MaaS and MaaR can be thought of in order of suitability for DARTHs to fit the paradigm of the DE concept. MaaA are good, for instance, for those who want to claim the intellectual property rights or for commercial environments but they are a bad thing for science. Their code is a block where independent revision is nearly impossible. Testing new features, which is per se a problematic issue (Kelly and Sanders, 2008), becomes almost unviable in MaaA. In fact, these models have been implemented as monolithic codes where the absence of separation of concerns makes it difficult to read/debug them (Serafin, 2019). If a modeller is interested in using a particular MaaA function, this is actually not possible and coders are re-implementing the same things over and over. MaaA are usually defined as "silos", in the sense that they cannot exchange data and procedures and do not favour the exchange of knowledge between related disciplines.

MaaT is a step forward that MaaA can gain after a robust refactoring. They preserve as benefits (from MaaA) a strong control of the model's use and execution. However, the limitations on the usability and flexibility of the model are evident. Within MaaT everything is controlled by the developers who not only establish how and when the models can be used but also control the model evolution and enhancement. Clearly if the provider is an organized community with rules for getting contributions, the model can be a "community model", in the sense, for instance, that CLM 5 (Lawrence et al., 2019) and Jules (Best et al., 2011) are.

MaaS interoperability consists instead of machine-to-machine interaction through a published API, e.g. for a run configuration and execution. Nevertheless, it is not possible to move the model and make it run on a different machine transparently without human intervention.

The previous three levels establish a dependence of the user on the provider of the modelling services. What we should aim for with DARTHs is to accomplish the more flexible possibilities represented by the MaaR. MaaR allows one to effectively move the model and make it run on the machine that performs best for a specific use case, with clear benefits in terms of

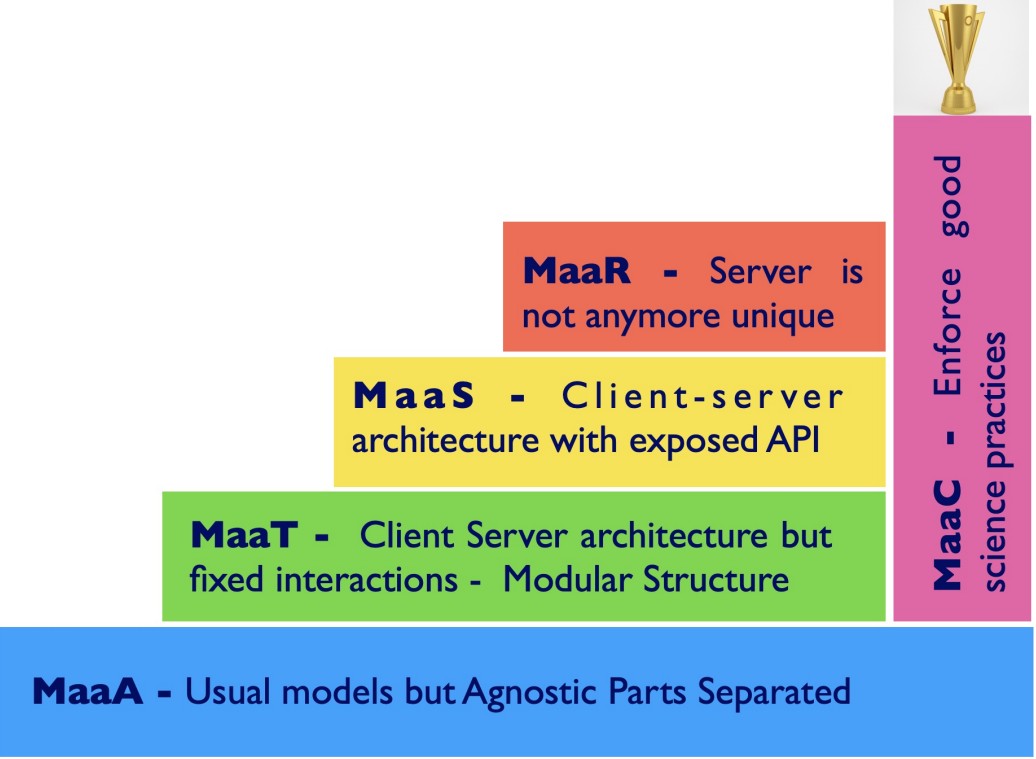

**Figure 3.** The convergence of present models to DARTHs aware models can be seen as a few steps that involve introducing some software layers, even starting from present modularized models (MaaS). They include the structuring of client-server architectures initially with a fixed client interface and server (MaaT), a client-server architecture where the connections can be obtained machine-to-machine by API (MaaS), and multiplicity of client and servers where models can be dispatched at run time according to the computational needs without intervention of the users (MaaR). At the same time, in order to fulfill the realistic and reliability requirements, a certain number of open science-aware models that fall under the name of MaaC should be pursued.

scalability, and interoperability. MaaR obviously requires a flexible infrastructure that allows models to be built rapidly and openly, and protocols to make the parts of the system work together without side effects. While fully-fledged MaaRs do not yet exist, for instance the Cloud Services Innovation Platform (CSIP) (David et al., 2014) and CSDMS Peckham et al. (2013a) are

something between MaaS and MaaR. However, so far, they have not been as widely adopted as they could be, indicating some possible complexity in their use that coders and runners could not face properly, or some missing action in disseminating their added value among scientists. In the era of big data, various frameworks can provide executions of software tasks over the web. For instance, Kubernetes (https://en.wikipedia.org/wiki/Kubernetes) is an orchestrator of containers that automate software deployment and management that could be used for some of these tasks. Airflow (https://airflow.apache.org/) is a manager of

workflows that could be arranged to manage these MaaS and MaaR software layers, even if DARTHs would require a more "declarative" (see the glossary) interactivity than the one that AIRFlow implements.

The informatics itself, however, does not guarantee the scientific contents of the models that can be run on the infrastructure. Therefore, we claim that a further, fifth step, has to be made, which covers both the science and technological sides, to obtain:

- Model-as-a-Commodity (MaaC):In the case models can be chosen from a pool present in the cloud, not only can they be automatically connected and made to run with a minimal intervention to respond to the needs of the users demands (as essentially required by MaaR) but they can also be contributed to, modified and expanded by the technical user to fit their purposes. Thus the pool increases as science advances. MaaC are also required, by construction, to embed tools for assessing their reliability, support literate computing, as supported for instance by Notebooks (Lau et al., 2020) (but not limited to them) and allow for hypothesis testing.

MaaC are demanding, as they should make the process of coupling models and the addition of features as automatic as possible, with the least input possible from the runner or coder. Actually, some conceptual work still needs to be done in this direction, as pointed out by the example of the infiltration model discussed in Peckham et al. (2013a). As shown in Figure 3, many of the MaaC requirements can be built in parallel to the infrastructures that implement the requirements of MaaT through to MaaR .

The listed characteristics of MaaC have implementation consequences that will be discussed in the following sections. A key feature of the most evolved modelling infrastructures is the possibility of breaking down models into parts that can be reassembled for a specific purpose, which all depends on the possibility of a modelling infrastructure to make model parts interact, and to allow them to be modified, evolved or changed. A mature DARTH requires MaaCs.

## 2.5 Back to core software engineering for models

Building models differently from what was traditionally conceived (Voinov and Shugart, 2013) is the way to achieve the flexibility required by DARTHs systems, which spans different methods, roles, uses and the needs to describe different resolutions and scales, and needs to consider new assumptions and paradigms, and extend their scientific domain over several traditional disciplines (Savenije and Hrachowitz, 2017; Bancheri et al., 2019b).

Unlike past practices, structuring the software into composable parts, called "components" should become the standard. Figure 4 shows the functioning of such a component, Prospero, that estimates evapotranspiration (ET). The Prospero component uses two other components: one describing the canopy and the other describing the constraints (stresses) limiting evaporation. Each component exchanges information with the other through their inputs and outputs, as indicated by the arrows in Figure 4. All internal components, per se, are not accessible at runtime (according to the information hiding principle -for its definition, please refer to the glossary) and the core algorithms of the components can actually be changed in favour of others that perform faster or more realistically without altering the overall scope of the system (i.e., estimating ET).

Components can be interchanged by the runners, chosen from a pool and linked together (in the case illustrated by using a script in a Domain Specific Language, DSL).

This modelling-by-component (MBC) approach has actually existed for more than forty years (Holling, 1978) but it is only in the last twenty years that it has gained momentum in the environmental modelling community (Argent, 2004). Often, it was

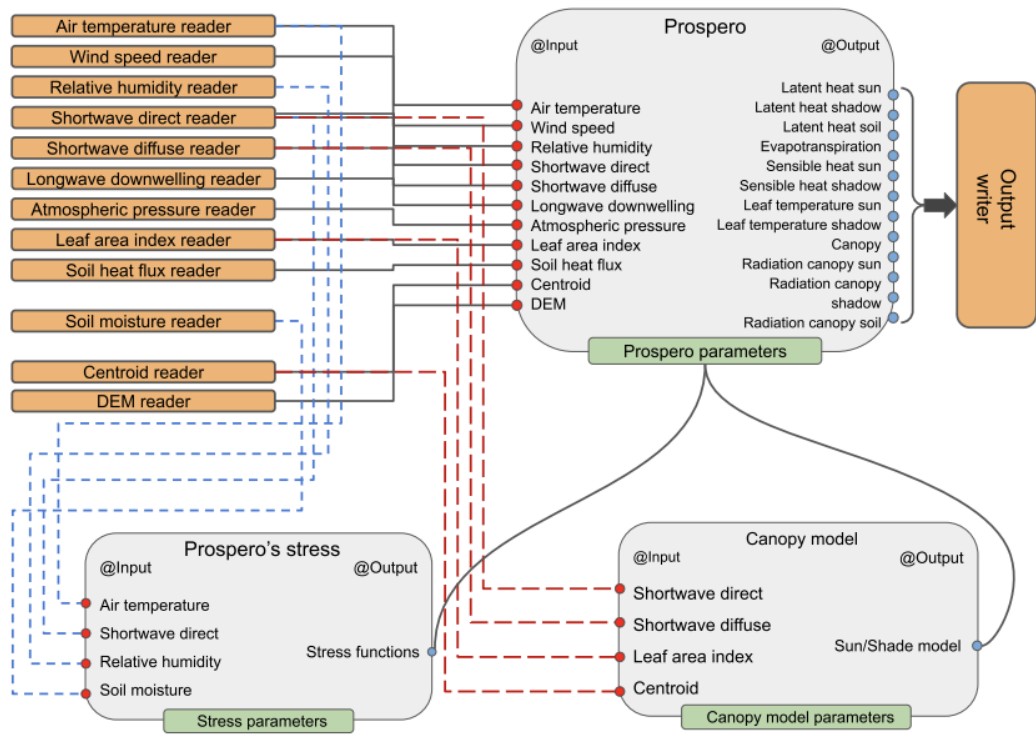

**Figure 4.** A visualization of models components. In this case, Prospero is a component of the GEOframe system (Bottazzi et al., 2021) that estimates evapotranspiration.

named "Integrated Environmental Modeling" (IEM) because it responds to the request of studying heterogeneous processes together and it integrates knowledge from various disciplines (Moore and Hughes, 2017).

MBC has just a few examples in the more restricted hydrological and meteorological community, some of which are, for instance, TIME (Rahman et al., 2003), OpenMI (Gregersen et al., 2007), CSDMS (Peckham et al., 2013b), ESMF (Collins et al., 2005), OMS (David et al., 2013). A longer list can be found in Chen et al. (2020).

MBC concepts and its technological consequences are tempting, but their real deployment can suffer from "invasiveness" in some cases (i.e. they may change the habits of a good programmer, e.g. Lloyd et al.,2011) and they require quite an adaptation of the usual programming styles. Among these, OMS faced this issue, explicitly finding encouraging solutions (Lloyd et al., 2011). MBC promotes server-oriented-architectures of the software (SOA), which is the same type of software architecture that was requested for treating the heterogeneous data sources. In principle, the SOA framework can work on different machines,

and be scalable across various hardware architectures. The final coder or user does not have to take care of the details of the computational engines because the framework itself takes care of it. Good features of MBC can be recognized to be:

- Encapsulation that simplifies code inspection. Components can work stand-alone (supported by a given infrastructure) and each of them can be tested separately;

- Well established intellectual property. Each component usually has a few contributors and different components involve a diversity of developers without being dispersed in thousands lines of code. Adding components is always possible and does not require a recompilation of the whole system;

- Substitution of components is easy and the use of components in hypothesis testing favoured (Beven, 2019);

- Basic services, such as implicit parallelism or tools for the calibration of model parameters, are provided under the hood, as explained in the next section;

- In a well designed system, the composition of modelling solutions is practically unlimited and components can accomplish a wide range of tasks, not necessarily from the same discipline (i.e., silos are avoided).

Therefore, this way of designing and organizing codes is the natural candidate for fulfilling the requirements of DARTHs and for providing the building blocks of futuristic MaaC infrastructures. However, MaaC has to be treated with care, since it can potentially lead to the spreading of unreliable or untested models. Because of this, MaaC should not be limited to 455  simply providing the results of simulations, but they should be equipped by design with a collection of tools for assessing the degree of reliability of the results and for quantifying their errors (uncertainty). This aspect requires further investigation in the technicality of MaaC.

## 2.6   How to write and manage models

If MBC is a necessary ingredient of DARTHs, then we have to understand how to write the code inside the components. The 460  first requirement is that the code has to be clear: in fact, in order to accomplish the open science ideas, it can be argued that not only the code has to be open source, but also understandable. Millman et al. (2018) offers an overview of good practices to be followed. Years ago, Donald Knuth proposed the concept of literate programming (Knuth, 1984). He proposed tools to enable users to integrate texts and figures with code, and then, with additional tools, to prepare documents in which the code is explained. The idea, however fascinating in principle, did not really gain momentum among programmers, even though there 465  were programming specific tools such as WEB (Knuth and Levy, 1983), Sweave (Leisch, 2002), Knitr (Xie, 2013) and others that would have allowed it.

It was Martin (2009), in fact, who proposed that the best documentation of the code is the code itself and its appropriate organization. With respect to Knuth's vision, some factors play a major role today, particularly the birth of high-level languages that allow a greater expressivity than was possible in the early years of high-level languages. In the early days, memory was so 470  tight that variables and other names were usually single letters, often reused from scope to scope, and looking at a code base

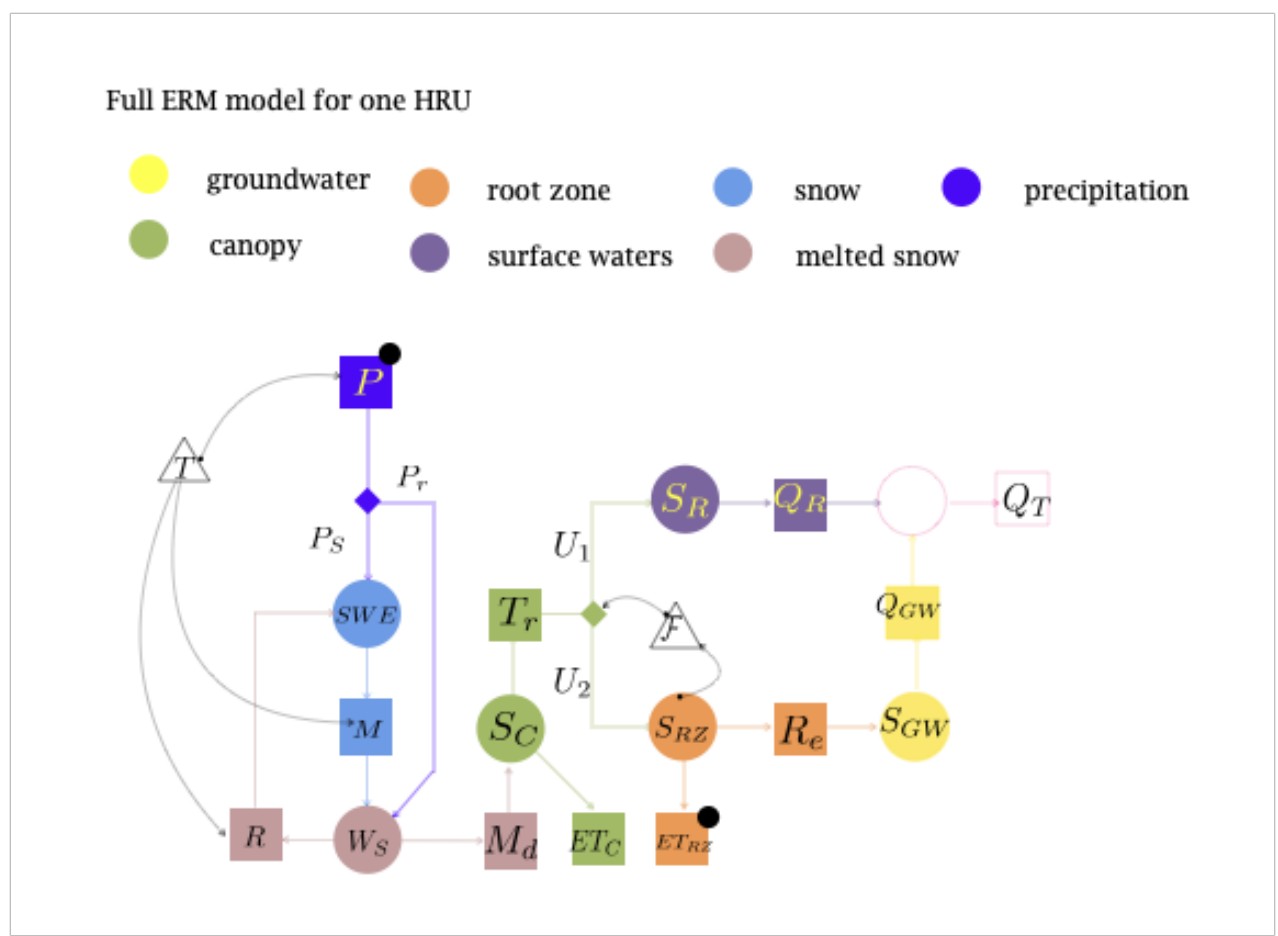

**Figure 5.** Representation of the Embedded Reservoir Model, Bancheri et al. (2019a), that is based on a system of six ordinary differential equations that represent the hydrological fluxes in a point. HRU stands for Hydrologic Response Units. See the text for further explanations.

was more decrypting than reading. Now, languages allow the use of more expressive names and most of the languages have standard ways of inserting comments (to the point of making them invasive sometimes), which can be processed to produce documentation. One of the clearest examples is the Javadoc tools (Kramer, 1999).

Another issue that can arise with open source codes is the organization of the classes, which is obscure most of the time, and
475 often what is gained in clarity by having classes with short code contents is lost by having hundreds of them without having a clue of their use in sequence. External documentation with UML diagrams can help this phase, but it is usually a neglected part of the documentation. Some languages, such as Java, have recently introduced the concept of modules that serve to specify the dependence of one part of the code on others. A further step on this topic is to adopt state-of-the-art building tools, like Maven (https://maven.apache.org/) or Gradle (https://gradle.org/), that collect all the dependencies needed to compile the codes and, if
well written, at least clarify the dependences at package level. These tools are much more evolved than the traditional "make"

command of the C language, since they allow one to grab codes from globally diffuse repositories and keep each building process up-to-date, together with the latest version of the libraries they use.

Going even more in depth into the programming, one aspect that is usually ignored by scientific programmers is that object oriented programming is not just about using classes but about organizing them efficiently as projects grow in dimension, as DARTH philosophy requires. These aspects are treated in "design patterns" (Gamma et al., 1995), which offer a mature understanding of what object oriented programming is although they are often not part of elementary books on programming. A few books indeed cover the issues for scientific computing, such as Rouson et al. (2014) for FORTRAN95 and Gardner and Manduchi (2007) for JAVA. Another step forward can be made by using a certain degree of abstraction in software, as proposed, for instance, by Berti (2000) with examples in C++ and used in Tubini and Rigon (2021) using JAVA. Abstraction, one of the basic rules of design pattern (i.e., " Program to interface and not to objects"), is important in that it allows the software "not to be modified" but "extended", thus enormously increasing the flexibility of a project and its lifetime. This software engineering technique enhances the flexibility already available by MBC and permits it to limit the code dimension and, when well engineered, a massive code reuse.

All of the above prescriptions can have different deployments in different languages but the overall principles are: reuse code, make it shorter, easy to read, and easy to extend while controlling modifications that can be disruptive to the stability of the whole.

Recently, a practice has been introduced by the Journal of Open Source Software, Joss LINK, to review the model codes as part of the peer review process. It is a possibility that DARTH coders should not overlook. It is clear, in fact, that the credibility of the results must be accompanied by the inspectability of any part of the research and, when this is based on extensive computing, it must include access to the code's internals.

To DARTH, the concept of literate computing (Rädle et al., 2017) can be more useful than literate programming itself. This is, for instance, promoted by the Mathematica Notebooks (https://www.wolfram.com/notebooks/) and, more recently, by the Jupyter Notebooks (Loizides and Schmidt, 2016) and others (Lau et al., 2020). One necessary characteristic of open science is to keep track of the setting of performed simulations and of the data manipulation used to infer results. Some frameworks, like OMS (David, 2013), have internal mechanisms to do this, by commanding the simulation by means of scripts that are subsequently archived in standard places with standard name extensions. However, Notebooks, with their mix of scripting, visualization of data and comments, greatly enhance the documentation of a creative scientific process, even if their role should not be overemphasized. For instance, the standard operative approach for GEOframe (Formetta et al., 2022 in preparation) users is to support the preparation of the analysis of inputs and outputs with Jupyter notebooks. Until more fancy ways of interfacing with models are available, these remain the best way to expose all that has been done to a scientific audience.

## 2.7   Assessing the reliability of DARTHs by design

An essential aspect of MaaC is the estimation of uncertainty in any type of prediction. This is essential from a practical point of view, i.e. for those activities that rely on model predictions for their planning activities, for the advancement of science, in the continuous activity between ideation, hypothesis testing and refinement. Furthermore, it is also essential for the viewer of

515 the results, including citizens, to understand the reliability of the results. According to our definition in the glossary, reliability is a relative concept, which itself requires studies, but models do have a degree of reliability if each of their estimations always comes with an estimation of its uncertainty.

We are taught that we cannot validate models but, with the feeble light of statistics, we can try to understand how confident we can be in a certain prediction. Götzinger and Bárdossy (2008) and Beven (2018) present a very clear summary of the
520 issues but there is a multitude of other contributions that treat the topic. Binley et al. (1991) and Todini (2007), among others, dissected the problem related to the forecasting of discharges, but any model of any type involves errors and has its own error producing mechanisms, as can be seen, for instance, in Yeh (1986) or Hill and Tiedeman (2007) for groundwaters, Vrugt and Neuman (2006) for vadose zone variables, Post et al. (2017) for ecohydrological modelling, Vrugt et al. (2001) for root uptake, and Yilmaz et al. (2010) for general watershed modelling.

Literature is often concerned with where the errors are. Uncertainty comes from errors in data, epistemic errors, errors in methods for parameter calibration, and heterogeneity, i.e. the variability of the domain to be described and modelled. The last example could be included in the epistemic error but citing it separately serves to stress the point. Part of the uncertainty certainly comes from ignorance of conditions that, being unknown yet necessary to completely define the mathematical problem, must be guessed.

Errors in measurements affect the procedure of calibration and cause the inference of incorrect model parameters. Errors in model structure reflect in wrong forecasts, which in turn cause biased comparisons between them and the models outputs.

Figure 6 portrays the life cycle of a model: circles represent data; boxes represent actions. Input, boundary and initial conditions can contain errors that propagate to the model results, leading to different responses than the control measurement set. Model results contain bias and variability generated by errors in the model structure, which also contains parameters or
535 hyperparameters, if the model is a machine learning one, to be determined. A phase of calibration/training follows to change the model's parameters by evaluating a metric of the difference between predictions and measurements (Goodness of Fit, GoF). During this phase the model is run with different sets of parameters until the optimal set of parameters is found, which correspond to the prescribed GoF value. When a satisfactory agreement is reached the model can go into production, or to a further level of analysis with stakeholders.

Once calibration of the model has been done to bring results up to a pre-established level of acceptability, a further evaluation of the model's results must be done to test the adequacy of the model to solve the problem it was intended to address (Refsgaard et al., 2007).

Usually, the decision-making process implies the existence of a group of stakeholders with a clear methodology, or at least a set of beliefs on which to base their final judgments. However, this is not necessarily the case of DARTHs, where in principle,
people can ask for model results without having an informed background. Therefore, even though the range of results that a model provides is restricted, given the laws of dynamics and thermodynamics, DARTHs should by design give appropriate warnings when results require discussion or interpretation. Among the myriad procedures for calibration, sensitivity analysis, and data assimilation, error estimation is more art than science: while the methods are rigorous, the assumptions under which they work are of variable credibility depending on the process. For instance, Refsgaard et al. (2007), recognizes at least 14

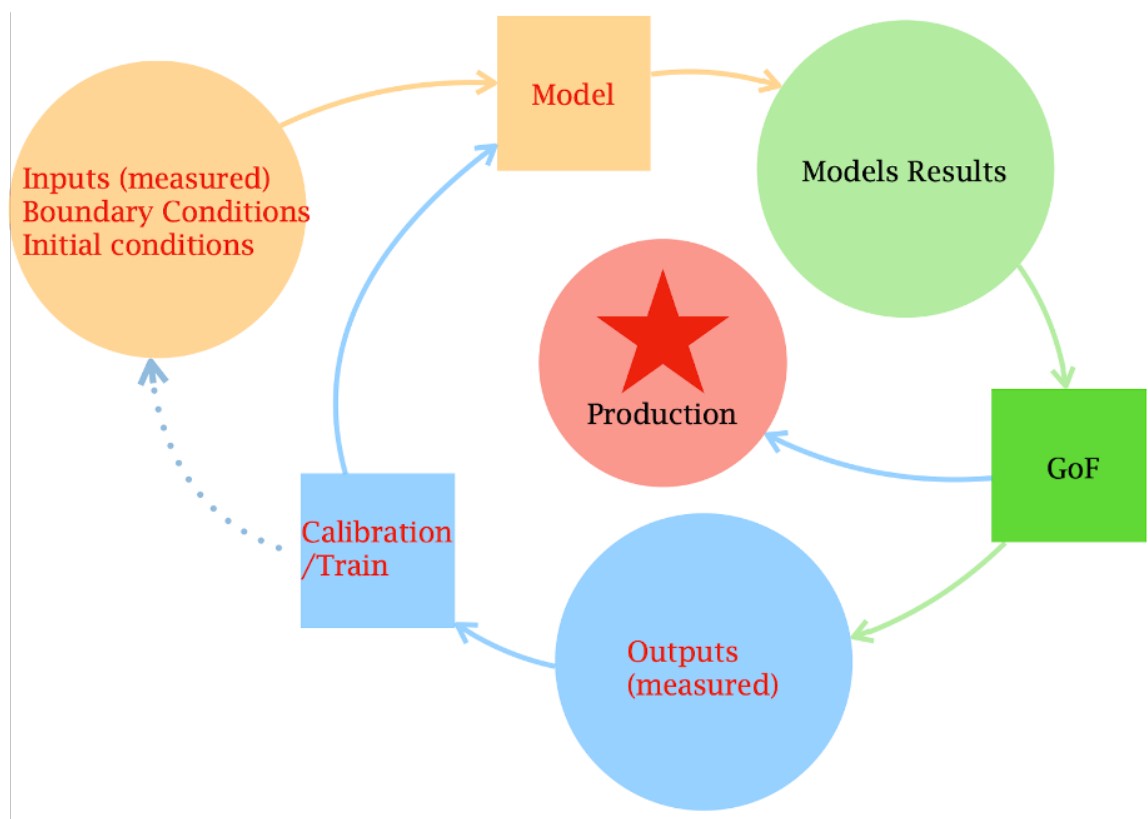

**Figure 6.** The life cycle of a model.

methodologies (plus GLUE) to obtain these estimates. Here, we do not advise the use of any particular one of these methods, but
we do claim that at least one method should be chosen. If, for the sake of the advancement of science, the search for the origin
of errors is paramount, for that which regards DARTHs we stay with the simplest fact: "purely empirically, probability and
statistics can, of course, describe anything from observations to model residuals, regardless of the actual source of uncertainty"
(Cox, 1946). Moreover, we want to reinforce the idea that error estimation is a practice that has to be continuously exerted and
refined. If the comparison between computed and measured quantities is systematically done in DARTHs, then statistics of the
performance of a certain model setup becomes more reliable with time. As a consequence, we can easily figure out what was
the success rate of a model's predictions and use it as a baseline for future decisions. At the same time, using experience and
comparison, we can improve the reliability of the methods used to assess uncertainties. That is to say, we can test and improve
both the results and the methods to determine their precision.

Model assessment occupies the space between parameter determination (Gupta et al., 1999), sensitivity analysis (Pianosi
et al., 2016), data assimilation (Reichle, 2008) and decision making (Refsgaard et al., 2007).

## 3  Enabling technologies and DARTHs

DARTHs are themselves an enabling technology. However, according to the recent developments, they can also contain new or relatively new "technologies". Those envisioned here are: high performance computing, which is actually a necessity for DARTHs; Earth Observations, which can satisfy the data voracity; and Machine Learning approaches to simulations. These technologies have entangled functionalities, as we describe below.

### 3.1  Dealing with the computational burden

A DARTH needs to be supported with the appropriate informatics that allow it to distribute parallel simulations and search with appropriate scalability among multi-core machines and cloud infrastructures; these are further requests for the HPC applied research.

The DARTH metaphor requires an extremely large use of data exchange in the background, which requires extremely high computational power.

Efficiency in a DARTH must be achieved in many ways. For instance, data gathering and ingestion: from the coder to the runner, it is important to have self-explanatory data formats that are efficient in memory (Lentner, 2019). DARTH action, as it has been envisioned so far, can be represented by a graph where the nodes are computational points, while the arrows represent potential exchanges of data. In Figure 4, we already presented what a chunk of a DARTH should be (without parallel routes). A more complete view is presented in Figure 5 where, according to the Extended Petri Nets (EPN) representation of hydrological dynamical systems (Bancheri et al., 2019b), the model is formed by the resolution of six ordinary differential equations (ODEs). In Figure 5, each ODE is represented by a circle, while the model fluxes are represented by squares. As the graph shows, there are at least two independent paths that produce (as a sum) the final discharge QT, and these paths can be run in parallel. If we write the informatics that solves the system of equations, each of the ODEs (i.e. the circles) is mapped to one or more DARTH components, and some other components are required to estimate the fluxes (i.e. the squares). For instance, the green Evapotranspiration flux, ETc, in Figure 5 corresponds to components represented in Figure 4. Therefore, stripping down the mathematics can reveal further possibilities for parallelization. In the hands of computer scientists this can be done and, for instance, it has been studied in the OMS framework (David et al., 2013, 2014). In fact, OMS/CSIP uses knowledge of the component connections to run them in parallel over multi-core machines or on cloud computing services.

Another aspect relevant to hydrologists is that the Earth's surface can be tiled into catchments and subcatchments (Rodríguez-Iturbe and Rinaldo, 2001). This organization of the space can be utilized to better organize the computation and reduce the information that needs to be passed between the areas. In Figure 7, 32 Hydrologic Response Units (HRUs) and two lakes are distinguished, and each one of them can be partially processed in parallel. For instance, the Net3 infrastructure (Serafin, 2019) has been deployed in the OMS core and serves this scope. It treats the external HRUs (shown in orange at the bottom of Figure 7) independently and then moves down the computation following the channel network. It is worth noting that, if the parallelism is obvious when two paths separate, such as in the graph in Figure 5, a pipeline type of parallelism (McCool et al., 2012) can also be implemented for the linear sequences of models represented by a tree-like graph. In a pipeline, one

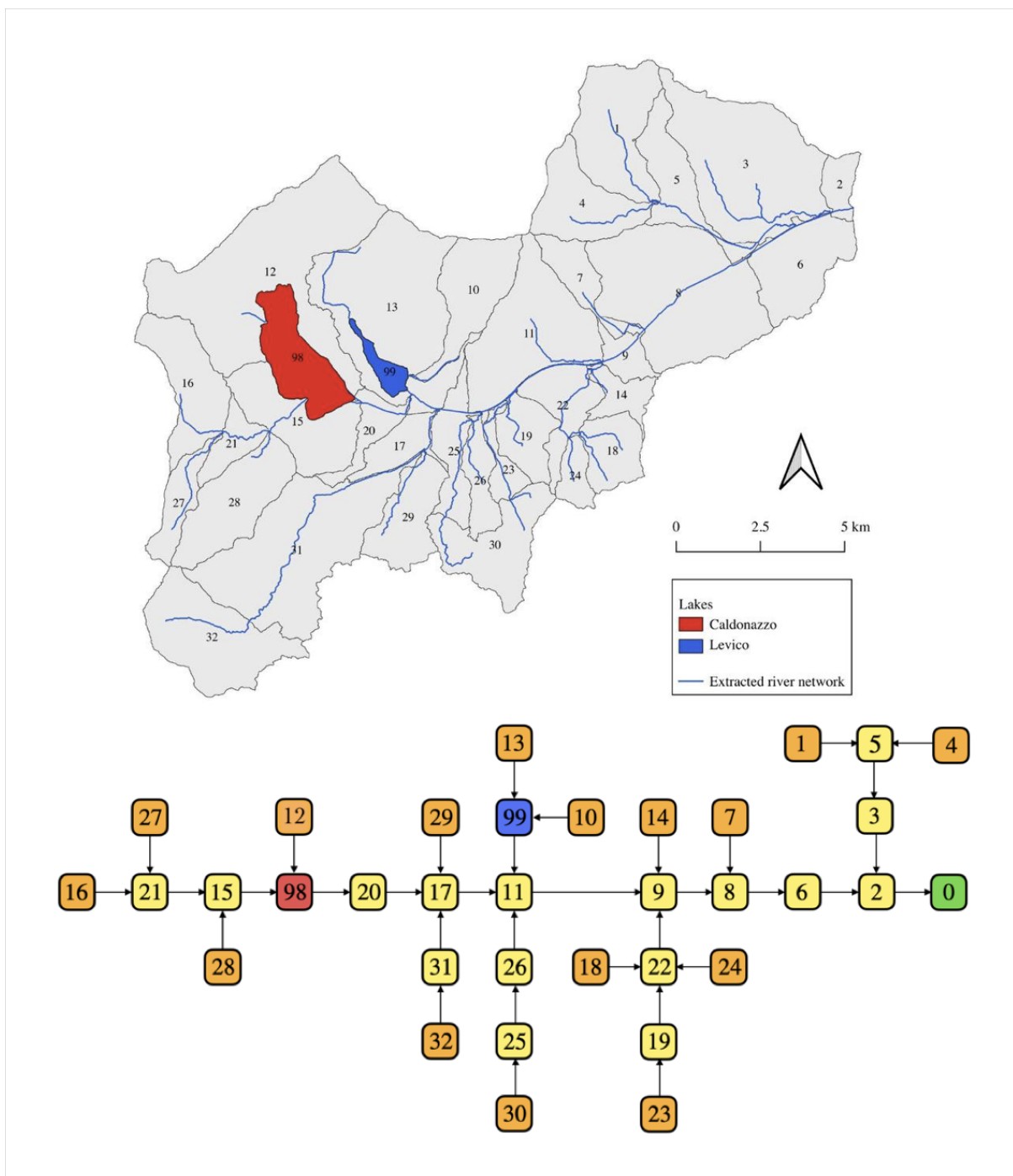

**Figure 7.** The upper river Brenta (Italy) is a catchment subdivided in 34 Hydrologic Response Units, including the two lakes of Caldonazzo and Levico. The channel network gives an ordered structure to the geographic region that can be used to process the HRU in parallel Modified from Busti (2021)

component gets the input from another; thus, when the graph is straight and acyclic, the work done by a process, represented by a node in the graph, can be performed by a group of processors or cores and, once one of these is freed up, it can be used again for a new job. These techniques are well known to computer scientists but usually not to hydrologists. In Figure 7, each square node corresponds to a system that is, in most cases, equal or similar to the one presented in Figure 5. Therefore, the underlying infrastructure has to harmonize the various levels of parallelization: the one at component level (multiple ones for

each node) with the one at the river network level. This actually happens, for example, in OMS and we refer to it as "implicit parallelization".

Implicit parallelization has the great advantage that its action is completely hidden to the coder who implements the components and its mechanism does not affect the way the coder implements the processes. If the framework developers find more efficient ways to achieve it, they can change the engine without the hydrological code necessarily being changed.

Purists of HPC could argue that these forms of computation are far from the efficiency of codes tuned in an assembler to obtain the maximum possible performances out of any hardware. However, here what matters is the coders' and runners' ease of use and implementation, as it can lead to the greatest improvements. In well designed DARTHs, coders can completely ignore the computational pains in parallelizing the codes and instead focus on the tasks they have to solve, i.e., to cook their ingredients instead of taking care of building the kitchen.

It is worth mentioning at this point, with an eye on participatory computation and programming, that the spatial partition of catchments offers the possibility of using different types of models in different locations. For instance, in Figure 7 the two lakes (blue and red nodes) are modelled with a different set of equations than the yellow and orange nodes. Moreover, the runs performed for the characterization of the subcatchments that are upstream of the lakes, for example, can be performed by different runners who can share just the final results of their work. To this scope, however, catchments (and their partitioning

HRU) need to have a unique identification, ultimately all over the world: but this is what latitude and longitude and their decimals were invented for.

One possible objection to what has been presented is that process based models (Fatichi et al., 2016b), in the traditional sense like Parflow (Kuffour et al., 2020), GEOtop (Endrizzi et al., 2013), Hydrogeosphere (Brunner and Simmons, 2012), and SHE (Refsgaard et al., 2010), which have outstanding numerics and capabilities, require a gridded partition of the Earth's surface that

does not seem to fit in the nested-graph structure just envisioned. This objection is only partially valid, however, since process-based models have functional parts that can be refactored to be loosely coupled in components, while the computational grids can themselves be advantageously disjointed into spatial parts and their computation organized in graph-like structure. Certain parts of the codes, though, could remain tied to the grid structure to which other forms of parallelism methods can be applied. For these, other strategies of computation can be envisioned, in which global computations, run by institutions, can be used to

drive local computations, similarly to what is done with meteorological models. The use of process-grid-based models, in fact, constitutes a third type of HPC demand that should be harmonized with the others. In summary, various levels of optimization of the computational resources can be activated in DARTHs, depending on modelling choices. However, as we have shown, some of them can remain transparent to the coders and the runners, simplifying their work so that they can be concerned with

just the hydrology/biogeochemical physics of the processes rather than with the informatics. All this upon the adoption of the right framework, which appears, at this point, to be twice necessary.

## 3.2 DARTHs as a bridge between Earth Sciences and Earth Observation

There is a pronounced trend in current hydrological research to use Earth Observation (EO) as the basis of the Digital Twin Earth (DTE). ESA and NASA have enthusiastically embraced the idea within their programmes (https://www.esa.int/ Applications/Observing_the_Earth/Working_towards_a_Digital_Twin_of_Earth). And, in lay people's imagination, the visualization of their resources on virtual globes is what comes closest to the idea of DE. From the standpoint of hydrological sciences, EO data, and specifically those derived from space-based sensors, have provided new and independent datasets that span the range of water cycle components (the reader is referred to McCabe et al. (2017b), Lettenmaier et al. (2015), Babaeian et al. (2019) for further details). The usefulness of Earth observations, however, lies not just in their capacity to reveal insights on the water cycle, but also in their potential to benchmark Earth system models. The latter is particularly important when dealing with the representation of human processes by models (Müller-Hansen et al., 2017) as it is undeniable that the interaction between human activities and the hydrological cycle is currently stronger than ever (Abbott et al., 2019; Wada et al., 2017).

Despite the huge availability and variety of data from EO, the extent to which current hydrological models can efficiently and effectively ingest such massive data volumes is still poor. In meteorological and atmospheric sciences the exploitation of EO data has been supported by the development of community-based models and Data Assimilation (DA) systems, like OpenDA (www.openda.org) and the Land Information System (LIS, Kumar et al. (2006); Peters-Lidard et al. (2007a)). On the other hand, in the hydrological community this has only partially happened. The reasons for this range from differences in the scale of applications (hydrologic hindcasting and forecasting has been more oriented to smaller spatial and temporal scales) to aspects of the underlying physical system (i.e. atmospheric vs. hydrological systems vs. ecosystems), with hydrological systems and ecosystems characterized by different, highly nonlinear models for a variety of processes which existing DA and EO techniques may be unable to handle (Liu et al., 2012; Kumar et al., 2015).

If both communities embrace the DARTH logic explained in the previous sections, the hydrological community should learn from the meteorological and atmospheric communities and accelerate the transition from fragmented hydrologic data assimilation research towards community-supported, open-source systems that can operate an efficient ingestion of EO data. The EO community can introduce more sophisticated retrieval processes and benefit from the granularity permitted by MBC and use the mass and energy conservation laws to better define what EO sensors see. Besides using various sensor data, current practices in deploying EO-based products also use hydrological assumptions, such as simple water balance schemes, and different formulations for losses, such as drainage and evaporation (Martens et al., 2016; Manfreda et al., 2014; Brocca et al., 2014). Then, as often happens, these EO-based products are used by hydrologists as forcing or calibration datasets in their hydrological models with the purpose of obtaining the discharges out of a catchment. The hydrological models, however, usually have a different structure with respect to the ones that the EO products are based on, resulting in potential spurious uncertainty and not always optimal results (López López et al., 2017). Instead the DA could be completely incorporated with the simulations without the need for the intermediate steps in which the EO community deploys its products and the hydrological

community its models, and the outcomes of both are compared at a final stage. Papers like Martens et al. (2016) and Meyer et al. (2019) can give guidance on how to achieve this.

Generally speaking and with few exceptions, geophysical variables that are derived from satellite data are obtained via complex retrieval models with numerous underlying assumptions, assumptions that are themselves different from those used by hydrological models. This workflow could be greatly simplified if model components could directly assimilate the spectral signatures of solar and Earth emitted radiation. To achieve this, DARTH should be equipped with model operator components (e.g, backscatter and radiative transfer models) that are able to use states, fluxes and ancillary information derived from EO

data to directly ingest brightness temperature and backscatter observations (De Lannoy and Reichle, 2016; Lievens et al., 2017; Modanesi et al., 2021). In this way DARTHs would represent the bridge between the hydrological and EO communities and would facilitate the participatory science we are envisioning here.

    Other challenges emerge when trying to integrate EO data with hydrological and land surface models. These are related to the mapping of observations to model variables, to the specification of model and observation errors, and to the homogeneity and harmonization of EO data (McCabe et al., 2017b). It is of common interest to solve the mismatches that have been

found between hydrological/land surface models and EO, for instance, when observing the spatial variability of soil moisture (Cornelissen et al., 2014; Bertoldi et al., 2014), or evaporation data (Trambauer et al., 2014). As with the tighter workflow just envisioned, the structure of conceptual hydrological models, which, for instance, could be obtained by producing hybrid PB+ML models as described in the next section, has to be improved. This, in turn, could bring to a rethinking of the processes

of calibration and DA, as envisioned by Tsai et al. (2021) and Geer (2021), albeit from different points of view.

    In any case, we suggest that the two communities – hydrologists and remote-sensing scientists – should have a stronger and closer collaboration, and DARTHs can provide a way to facilitate it. With DARTHs, we should not have hydrologists using satellite data simply as end-users, without giving feedback to remote-sensing scientists; and vice-versa, remote-sensing scientists should take care of the suggestions and criticism made by hydrologists.

## 3.3   The grand challenge of hybrid models in DARTHs

We have been agnostic, so far, with respect to type of models, whether DPB, HDS, MS or ML; getting into the details of these, their achievements or limitations, is not the argument of this paper. However, as Nearing et al. (2021) emphasizes, there are some questions that hydrological research is struggling to find answers to. Among these, for instance, one of the 23 questions posed by Blöschl et al. (2019): what are the hydrologic laws at the catchment scale and how do they change with scale? As

Nearing et al. (2021) points out, the elusiveness of answers to this question in the last 30 years was not caused by the lack of data, nor by the heterogeneity of catchments but more probably by some weaknesses of the physical-mathematical methods available (see also Gharari et al. (2021) for further discussions); it is necessary to work towards a new mathematical-statistical approach (Ramadhan et al., 2020). DARTHs should be implemented to respond to this request.

    The change of paradigm that is now expected will bring the possibility of merging different families of models within

695 the MBC informatics, and of hybrid modelling solutions (Shen et al., 2018) where MS, ML, HDS and PB can be mixed. ML techniques are current practice when using PB or HDS models, as for instance in the calibration phases where genetic

algorithms have been used since Vrugt et al. (2003), or particle swarm (Kennedy and Eberhart, 1997), but these techniques are not normally used in core modelling. Conversely, the community is more oriented to assimilate EO and it has had an articulated approach to ML (Reichstein et al., 2019). As previously mentioned, there is cross-fertilization between the two communities, but often diversity of objectives and mismatches of spatial and temporal scales (small scale vs large, global ones, hourly vs daily or larger time scales) generate misunderstandings and imprecise claims on the performances of the respective techniques. These operative and semantic gaps can be filled with the help of DARTHs.

The hydrological community is still learning how to use ML and there is a lot of room to incorporate statistical knowledge in PB models. As Konapala et al. (2020) writes, ML tools are used together with PB modelling, for instance, for calibration (Krapu et al., 2019), downscaling of hydrologic data (Abbaszadeh et al., 2019), rainfall-runoff modelling (Kratzert et al., 2018), data retrieval from remote sensing data (Karpatne et al., 2016; Ross et al., 2019; Cho et al., 2019), and interpreting the hydrologic process (Jiang and Kumar, 2019; Konapala and Mishra, 2020). Likewise, convolutional neural networks (CNN) have been used extensively to learn from spatially distributed fields (Kreyenberg et al., 2019; Liu and Wu, 2016; Pan et al., 2019). In most of these trials, PB calibration is performed first and subsequently refined with ML techniques. Obviously, in the production phase the same workflow is maintained. However the granularity of the mixing of models can go deeper: as in Geer (2021), we can use solvers in which some equations are normal PDE or ODE, and some others, simultaneously solved, are ML based like LSTM (Long Term Short Term Models).

However, there are a few technical issues that have to be resolved to satisfy the requirements of a well engineered DARTH. The solutions of the ML community were generated with a variety of problems in mind, like computer vision applications, speech, and natural language processing. As such, the modelling chains were developed somewhat differently from those traditionally used in hydrology and Earth sciences (Geer, 2021). These tools were created under the modelling assumption that ML can provide an all-purpose, non-linear, function-fitting capability, that is 'a universal approximator', as said by Hornik (1991). This approach, in fact, is not very different from the idea that PDE or ODE are the only tools required to solve any dynamical problem and that they are, from one perspective, like the openFOAM (Jasak et al., 2007) or the Fenics (Blechta et al., 2015) libraries that provide prepackaged solvers and methods for the most common equations types. These frameworks can be assembled before runtime by means of a domain specific language (DSL) and save time for researchers who, then, act as linkers. For instance in the case of neural networks, (Mayer and Jacobsen, 2020) ML frameworks supply libraries to define network types like Convolution Neural Networks (CNN) or Recurrent Neural Networks (RNN). As with openFOAM and Fenics, they provide common model architectures and interfaces via popular programming languages such as Python, C, C++, and Scala. However, unlike openFOAM and Fenics, a lot of the ML supports special hardware for calculation acceleration, with libraries such as CUDA Deep Neural Network library (cuDNN), NVIDIA Collective Communications Library (NCCL), and cuBLAS (the GPU implementation of BLAS libraries) for GPU accelerated deep learning training. Mayer and Jacobsen (2020) is a nice review of the ML frameworks currently available. Among these, we can mention pyTORCH (https://pytorch.org/), TensorFlow (Abadi et al., 2016), ML4J (https://github.com/ml4jarchive), H2O (Cook, 2016). Besides, these can use other frameworks, like Hadoop or Spark, to distribute calculations over heterogeneous cloud systems.The HPC computational forms of current PB and ML are also different. Earth science applications tend to use supercomputers, as weather and climate models, or PB like Parflow

(Kuffour et al., 2020), have a lot of inter-process communication (being grid-based) that is optimized on supercomputers. ML approaches typically use cloud computing, taking advantage of algorithms that require less communication along with the compatibility of neural network processing with graphics or tensor processing units (GPUs or TPUs). One practical problem, as highlighted in the previous section, is to combine typical MaaA supercomputer-friendly models with ML softwares.

In principle when using MBC, it would be conceptually easy to say that, in a given modelling solution, some of the components can be PB and some others ML based, and the workflow envisioned in the cited papers be implemented. For instance, (Serafin et al., 2021), has shown that it can be done by adopting an Artificial Neural Network (ANN) model for runoff purposes, in the OMS/CSIP system. In this case, the ANN libraries were integrated into the framework and this made the integration easier, though not trivial, but in general a more loose coupling through "Bridges" should be envisioned. This can be done, and one example is the Fortran-Keras Bridge (Ott et al., 2020) that allows a two way connection of MaaT to MaaC type of models with the Keras framework. Fortunately, the ML types of infrastructure are often engineered as SOA softwares and, therefore, it is not overwhelming to obtain the interactivity required by DARTHs within the flexibility offered by MBC.

One further issue is that there is not just one ML framework, but many. In ML, computations are driven by an internal graph structure, (Geer, 2021), but in some frameworks this graph is static, e.g., TensorFlow and Caffe2, while in others, for instance PyTorch, a dynamic graph is used. Therefore, the choice of the computation model can lead to some differences in programming and runtime. As such, this requires the implementation of a "translator" of the DL structures used in one framework into the others. At least one initiative has been born to accomplish this task, the ONNX standard (Lin et al., 2019), but it is missing an action, like the one offered by the OGC, to pursue that interoperability that DE and DARTH would require. Ultimately, because both PB and ML require computations on Direct Acyclic Graphs (DAG), it is possible to think that new common techniques of calculation can be shared by ML and PB models.

For the emerging world of HM to be fully operational within DARTHs three efforts are required:

– Building the appropriate Bridges;

– Working for unified standards of representation of ML processes, at least at the production stage;

– A possible re-engineering of the infrastructures to allow a finer granularity of the hybrid process modelling interactions.

## 4 Epilogue

How can we organize the building of DARTHs and which messages can we take home?

### 4.1 On the organization of DARTH communities and their governance

What we have said in the paper should suggest that the goal is not to build a fully fledged DARTH but, rather, to build DARTH services or components that can be assembled together to build a DARTHs solution. Moreover, various DARTHs can share some services and develop others for themselves, such as GDAL libraries that are used by most of the Open Source GIS systems or the Lapack libraries that are widely present in modelling infrastructures. We can think of DARTH components as

libraries but more shareable on the web and, in any case, as pieces of software that are self-contained and possibly can work alone. Therefore, the organization necessary for the development, maintenance and hosting of such a system, is dependent on which DARTH components we are thinking about. Since, for instance, EO data are provided by institutions like JAXA or ISRO, it will be their natural duty to provide the data in standardized formats and make them available in appropriate ways (i.e. by exposing an Application Programming Interface, API) to be linked into a DARTH solution. Other subjects can obviously provide their own elaborations or formats of the same data and make them available. For instance the eWaterCycle application (Hut et al., 2021) uses the common format CMOR and the ESMVARtool (Righi et al., 2020) can be used to make uniform otherwise heterogeneous data. If the core of DARTHs are DARTH components, provided by different institutions or companies, they should be eventually linked together to have a functioning DARTHs solution. Deployments like those initiated with eWaterCycle or Deltares-FEWS (Werner et al., 2013) and LIS (Peters-Lidard et al., 2007b) are examples of integrators of resources; maybe they are not fully compliant with the DARTHs architecture but they can be a good starting point to deploy DARTHs and provide an example of DARTHs providers.

Further ideas are expressed in Nativi et al. (2021), whcic has a specific section with a clear title: "Effective Governance of the Independent Enterprise Systems of a Digital Ecosystem" The word ecosystem, in fact, is probably the more appropriate in this context: not applications but ecosystems of open applications.

Because DARTHs grow and develop continuously by adapting to the emerging scientific questions, in principle there is no need for a centralized director for most of the developments. However, organized communities can achieve more. In particular, any community will be effective if it chooses to adopt common standards for data and formats, while also promoting innovative solutions. The Jules project (https://jules.jchmr.org/) is an example of governance for such communities. Examples of successful communities can be seen also in Archfield et al. (2015). An obvious and bigger model to learn from, outside the Hydrology community , is provided by the GNU/Linux project where one community is responsible for the kernel of the operating system and others, including GNU, provide additional tools. And at the end of the process other subjects deploy variously flavoured versions of GNU/Linux, called Linux distributions. The DARTH need not be either commercial or non-commercial: both strategies can be pursued (and, we guess, will be pursued). Please observe, however, that commercial is not the opposite of Open Source. An open source product can be commercialized and a commercial product can be open source. So DARTHs can be both commercial or non-commercial but, either way, would require appropriate open characteristics.

Building pieces of a DARTHs (i.e. DARTH components), as we see it, is feasible by an organized group of researchers (especially at the Linker level), or by partnerships funded by international research programmes. The OpenEO initiative, https://openeo.org, in the field of Earth observation data development is an example of how a DARTH development could be funded. The entire scope of this paper is to suggest a modular architecture with science parts that can, in principle, be affordable to many if not all scientists, as it was for JULES (Best et al., 2011), CLM (Lawrence et al., 2019), and other projects. From a lower level , SUMMA (Clark et al., 2015), GEOframe (Formetta et al., 2014) and OMS3 are efforts of a single research group that can provide a starting platform for a DARTH. Certainly producing a DARTH from scratch would be a major effort that only large company associations or governments could afford.

The quality control of the final product is done by, in fact it is, the community that builds a specific part of the DARTHs and the community or the commercial entity that links the various DARTHs components to form a DARTH distribution. Certainly, new techniques like the block-chain can be used to certify the steps and the chain of responsibility that a certain DARTH distribution has followed. It is expected that DARTHs initiatives will also come up with test cases with specific links to (or support from) important experimental activities, sites and observatories (e.g., https://www.hymex.fr/liaise/index.html) that link experimental measurements and observations to model development. The same observatories can also be responsible for building shared benchmarking datasets to test DARTH performance. If we refer to studies pursued with DARTHs, the ultimate responsibility is of the Runners who perform the studies and, eventually, the journal that publishes them. One critical aspect though, will be the availability of data, simulations records and software to allow the testing of results by other researchers. Whoever uses a DARTH is responsible for its results, just as whoever publishes something is responsible for the published piece. Experience from the SARS-COVID2 pandemic suggests that the reader must be educated and that keeping information hidden does not solve issues and does not cultivate the credibility of science. Therefore, in our opinion, limitations on the flow of information is a bad policy. On the other hand, the presence of errors estimation, the replicability of the experiment, and open code are landmarks for the good will of acting fairly.

## 4.2 Conclusions

In this paper we have discussed what DARTHs are and we found that they cannot simply be models as intended in the usual sense. They first need to be supported by an infrastructure that provides:

- the possibility to use the Modelling by components (MBC) strategy;

- implicit parallelism for simulation that mixes various types of parallelism;

- HPC to treat the data in input and output:

- loose coupling of models and data;

- HPC to perform calibration and data assimilation (DA) of parameters and retrieved quantities.

Besides, we claim that this infrastructure should be able to:

- dispatch data and models around the web for distributed elaboration;

- allow incremental improvements of the core programming features by a community of coders.

Detailed prescriptions were given for these infrastructures to accomplish open science requests, including being open source and produced by open source tools, a requirement that is, obviously, an option but that we think necessary for the progress of science. We also gave indications about deploying the necessary code and computing literacy.

We did not discuss the contents or the structure of the models but claimed that:

- DARTHs should be agnostic with respect to the model choice in order to support the progress of science;

– DARTHs should favour multiple hypothesis testing, which was fully discussed in many recent contributions (Clark et al., 2011a; Prieto et al., 2021; Fenicia and Kavetski, 2021).

Besides, to help progress, we categorized the models according to their characteristics in being able to cope with the DARTHs metaphor, spanning from MaaA to MaaC.

Those who are interested in model content can easily refer to the various commentaries, reflections and blueprints present in literature (Freeze and Harlan, 1969; Beven, 2002; Lee et al., 2005; Rigon et al., 2006; Montanari and Koutsoyiannis, 2012; Clark et al., 2015; Shen et al., 2018; Savenije and Hrachowitz, 2017), which should be thought of as complementary to the present paper.

We also remarked that DARTH has to encourage a new collaborative approach to EO for both the Hydrology (and Earth Science) community and the Earth observations community; an approach in which data fusion and products are made at the coder level and not provided as is to researchers. Another characteristic that has been invoked is the presence of readily available error estimates with any DARTH in any forecasting and as part of the knowledge creation.

A DARTH should not be thought of as an immutable piece of software but as a dynamically growing and evolving system in which different model paradigms, data and simulations can be exchanged and/or accessed over the web. Semantic information has to be added to allow searches of the competing tools and to make the discovery of data easier. A DARTH system should be openly managed and contributed to by the good will of researchers working within shared policies. DARTH design must favour cross-fertilization of knowledge between related sectors of science, technology and management, avoiding so-called silo-type models. The systems deployment concepts are, at present, themselves in development and they are a matter of research, trial and error. Therefore, DARTHs can be designed having in mind that many changes will be possible before getting the desired result.

We did not discuss the most commonly treated area of DE, i.e. the one dealing with visualization of data. It is obviously a fascinating subject, however, our previous experience in building GIS systems has taught us that visualization has to be treated as data and, therefore, loosely coupled to models because it is subject to rapid obsolescence. Many new human-computer interfaces are foreseen in the next few years and DARTHs have to be prepared to connect to them and exploit their capabilities, including the interfaces with the natural languages and body gestures that often appear in sci-fi movies.

Finally, there is an exciting area in HPC research where new methods can be envisioned for processing a chain of models and submodels along DAG that has just started being explored and that could unify the way the informatics and numerics of ML and MBC are actually done.

In this paper we gave a name to the hydrological DTE and expressed optimism that the goal to build a DARTH is a feasible enterprise that can be pursued with success. Many of the aspects touched upon are, in fact, already reality in current state-of-the-art practices and just need to be evolved, systematized, and integrated into a unique framework or made interoperable. This integration is necessary because, as the Authors believe, great advancements and innovations in hydrology can only be achieved by treating all aspects discussed together.

**Appendix A: Table of Acronyms**

*Author contributions.* The first Author had the idea of discussing the production of DT in Hydrology. All the Authors contributed creatively and equally likely to all the phases of work.

*Competing interests.* There are no competing interests

*Acknowledgements.* This paper has been partially supported by MIUR Project (PRIN 2017) "WATer mixing in the critical ZONe: obser-
865 vations and predictions under environmental changes-WATZON" (project code: 2017SL7ABC), the "Carbon and water cycles interactions during drought and their impact on WAter and ForEst Resources in the Mediterranean region" "WAFER" @CNR (Consiglio Nazionale delle Ricerche) and the ESA Digital Twin Earth Hydrology (ESA Contract 4000129870/19/I-NB)). We thank the two reviewers, Dr. Mark Hrachowitz and Dr. Uwe Ehret, and the Editor Dr. Erwin Zehe whose careful review helped to greatly improve the manuscript. English was revised by Eng. Joseph Tomasi.

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

**Table A1.** This table lists the main acronyms used in the paper.

| Acronymn | Meaning |
| --- | --- |
| ANN | Artificial Neural Networks |
| CEOS | Committee on Earth Observation Satellites |
| CNN | Convolutional Neural Networks |
| CSIP | Cloud Services Innovation Platform |
| DA | Data assimilation |
| DAG | Directly Acyclic Graphs |
| DARTH | Digital eArth Twin Hydrology |
| DE | Domain Earth |
| DSL | Domain Specific Language |
| DT | Digital Twin |
| DTE | Digital Twin Earth |
| EO | Earth Observations |
| EPN | Extending Petri Net |
| GEMS | Global Water Monitoring System |
| GEOSS | Global Earth  Observation  System |
| GRDC | Global Runoff Data Center |
| HDS | Reservoir Type models |
| HM | Hydrological Modelling |
| IEM | Integrated Environmental Modeling |
| IJDE | International Journal of Digital Earth |
| ISDE | International Society for Digital Earth |
| MaaA | Model-as-a-Application |
| MaaC | Model-as-a-Commodity |
| MaaR | Model-as-a-Resource |
| MaaS | Model-as-a-Service |
| MaaT | Model-as-a-Tool |
| MBC | Modelling-by-component |
| ML | Machine learning |
| OMS | Object Modelling System |
| PB | Process Based Models |
| RNN | Recurrent Neural Networks |
| SDI | Spatial Data Infrastructure |
| SM | Statistical models |
| SOA | Serve-oriented-architectures |