# Peer review of "HESS Opinions. Participatory Digital Earth Twin Hydrology systems (DARTHs) for everyone: a blueprint for hydrologists"

_Hydrology and Earth System Sciences, 2021_

## Referee Comment (RC2)

**'Participatory Digital Earth Twin Hydrology systems (DARTHs) for everyone: a blueprint for hydrologists'**

**by R. Rigon et al.**

Dear Editor, dear Authors,

I have reviewed the aforementioned work. My conclusions and comments are as follows:

**1. Scope**

The article is well within the scope of HESS.

**2. Summary**

In their opinion article, the authors describe both the necessity and the key aspects of moving hydrological modeling from their current state towards Participatory Digital Earth Twin Hydrology Systems (DARTHs). Key aspects of such systems include i) flexible coupling to many sources of data available in standardized formats , ii) platform- and language-independent implementation, iii) modular design facilitating recombination of model components. Each such aspect is explained in detail in a separate section and briefly summarized in the conclusions.

**3. Evaluation**

Overall, this article is a relevant and timely contribution to the discussion about the future of earth system modeling in general and hydrological modeling in particular. It discusses many current hindrances towards efficient, integrated and collaborative modeling and names key elements of a framework to overcome these hindrances. So I welcome the article for its content, but I do have several suggestions to improve its presentation:

- Key messages: For a reader not strongly familiar with the current state of development of Digital Earth (DE) and Digital Twin Earth Models (DT), only after reading the entire manuscript it becomes clear what the differences to current hydrological modeling practice are, and consequently what the authors aim at with their article. A reader will be able to follow the arguments in the paper much better if this is made clear from the beginning. I therefore suggest the following changes to the manuscript:

  - Include in the introduction a clear definition (purpose, key structural and organizational elements) of DE's, DT's, DARTH's and current hydrological models. Make clear where the latter fall short of qualifying as DARTHs, why it is necessary to elevate them towards becoming DARTHs, and what it takes to do so. A comparative list, or an explanatory chart can support this (maybe directly pointing to the sections in which the individual topics will be discussed in detail). From such an introduction which provides the overall picture, it will be easier for the reader to connect to the individual sections.

  - At the beginning of each topical section (sections 2-9), include a very short overview of the section content.

- Abstract: From reading the abstract only, I did not have an idea about what to expect from the article, and what its key messages are. For example, I did not know how to interpret the term "model as commodity", which is not standard terminology in hydrology. Please rewrite, taking into consideration my previous comment about key messages.

- L271: I partly agree to the statement "The peer review process is ineffective at sniffing out poor model validation", but while it may be far from perfect, I'd say it is still the best we have. Please

explain how in a DE/DT/DARTH world quality standards are maintained better than by peer control.

- Section 7: The section header just mentions "reliability", but in the text more aspects of model performance are discussed. I suggest widening the section header, and discussing robustness, reliability, reproducibility and realism (as mentioned in L260).
- Section 10: The conclusions are written in a rather hasty and bullet-point style, which devaluates the otherwise interesting and comprehensive paper. Please spend some more effort in writing a coherent and standalone summary and conclusion.
- Appendix A: Reading Appendix A, it was not clear to me why this section was moved to the appendix rather than being another section in the main text. I'd say the topic of proper coding is just as relevant for developing DARTHs as the other topics in sections 2-9. I suggest moving it to the main text.

Minor points
- L72: What is meant by "reasonable colour maps"?
- L82-84: It is unclear at this point what is meant by "tight, black-box models", or rather how they differ from the previously mentioned models. Please clarify.
- L89-90: The meaning of the sentence ("At present, …") is not clear to me. What is meant by "modeling panorama" here? Please clarify.
- L203: For me, the user's names are unfortunately not self-explanatory. Please add a short explanation similar to the explanation of roles.
- L353: Please explain the "information hiding principle", because it seems to contradict the general paradigm of openness and accessibility advocated throughout the paper.
- Caption of Fig. 5: What is meant by "IT"? Also, please add the channel network to the upper map for better connection to the lower map.
- Table B1 is very helpful. Please mention it already in the introduction, such that the reader can make use of it while reading the paper.
- Overall, there are many typos in the manuscript. Please check and remove.

I have another question to the authors related to current hydrological modeling and hydrological modeling in a DARTH environment. There is no need to address this in the manuscript, but I am interested in the authors opinion. The question is related to emergence as larger-scale phenomena arising from strong interactions on smaller scales. Current models build for a specific purpose and a specific scale/resolution often either make implicit use of emergence (by taking emergence as a given, directly representing the emerged phenomenon) or, because it may be relevant, setting the model up in a way that emergence may actually happen as the model runs (e.g. emergence of convective thunderstorm cells in convection-permitting atmospheric models). The latter often requires a well-attuned choice of time-stepping, spatial resolution, numerical scheme, variable precision, processes represented, etc. With a free and modular combination of model components in a DARTH environment, we may miss such well-attuned combinations, which may lead to not only incrementally but fundamentally different model behavior. How can this be taken care of in DARTH environments?

Yours sincerely,
Uwe Ehret

---

## Author Comment (AC1)

**Answer to Comments to the Authors by Reviewers of "HESS Opinions: Participatory Digital Earth Twin Hydrology systems (DARTHs) for everyone: a blueprint for hydrologists" by Rigon et al.**

**Answers to Reviewer Markus Hrachowitz**

**C 1** - I have really enjoyed reading this manuscript, although I do not necessarily agree with all of the points made. As a very welcome but (unfortunately) rare case, this Opinion piece develops a broad visionary perspective of what could be a very valuable step for the development of scientific hydrology as a core part of the Earth System Sciences. The authors do not only formulate their vision as a mere "wish list" but they also attempt to provide an outline of necessary major steps to be considered and potential challenges to be met along the way.

*A 1 - We thank the reviewer for the positive assessment of our paper. We are glad that Dr. Hrachowitz agreed with our (to use his words) "visionary perspective" towards a comprehensive way to do hydrology and hydrological modeling.*

**C 2** - I have a few observations and suggestions the authors may want to consider, as I think they may be helpful to strengthen the impact of their work.

This manuscript has been submitted to a hydrology journal. I therefore assume that the target audience envisaged by the authors are hydrologists and scientists/engineers from related fields.

As such, I suspect that many of our colleagues including myself may not be in detail familiar with some of the very technical and detailed computer science jargon used in the manuscript (some of them will be, of course) – in particular in sections 5 and 6. This may potentially also limit the level of appreciation and impact of this work. This would be unfortunate, really. I think there are two alternative ways of dealing with that issue. Either, the authors rework these heavy-jargon parts to make their language more accessible to a wider audience. Or the authors invest a bit more effort in more detailed explanation of the jargon terms, to allow the average reader to better follow their argument.

*A 2 - Thanks for the suggestion. In the revised manuscript we will adopt both of the strategies to reach a broader audience of readers. In particular, where the jargon is unnecessary, we will eliminate it in the revised manuscript. Besides we will add a short glossary for those terms which serve as a bridge between the different communities that have to work together in building DARTHs*

**C 3** - The authors cover the aspect of technical steps and challenges in a very exhaustive way. This is very welcome and necessary. However, I also believe that the vision for the development of DARTHs can be further strengthened by outlining some of the questions,

steps and challenges that will arise from an organizational perspective. This could include questions such as:

- Which type of organization is necessary for the development, hosting and maintenance of such a system?
- How can a decision process in the (further) development of DARTHs be designed? Who decides what? Can/should DARTHs be non-commercial or does it need to be designed as a commercial endeavor?
- Or in other words: who can afford it, how can financing look like?
- Who is responsible for quality control of items added by users?
- Who is responsible for avoiding misuse and misinterpretation of the models/data by non-specialists (e.g players, viewers and to some extent perhaps also runners)?

See also some aspects in Weiler and Beven (2015) form the perspective of a community model.

*A 3 - What we said in the paper should suggest that the goal is not to build a full-fledged DARTH but building DARTHs services or components that can be assembled together to build a DARTHs solution.* *This part was not emphasized enough in the original manuscript and we will modify it accordingly.*

*Moreover, various DARTHs can share some services and develop others for themselves such as GDAL libraries that are used by most of the Open Source GIS systems or the Lapack libraries that are widely present in modeling infrastructures. We can think to DARTHs components like libraries but more shareable on the web and, in any case, self-contained and possibly working alone pieces of software. Here below, we try to answer the above questions one by one.*

- C 3.1- Which type of organization is necessary for the development, hosting and maintenance of such a system?

*A 3.1 - The answer is dependent on which DARTH components we are thinking about. Since, for instance, EO data are provided by institutions like JAXA or ISRO, it will be their natural duty to provide the data in standardized formats and make them available in appropriate ways (i.e exposing an Application Programming Interface, API) to be linked into a DARTH solution. Other subjects can obviously provide their own elaborations or formats of the same data and make them available. For instance the eWaterCycle application (Hut et al., 2021) uses the common format CMOR and the ESMVARtool (Righi, 2020) can be used to uniform otherwise heterogeneous data.*

*If the core of DARTHs are DARTHs components, provided by different institutions or companies, they should be eventually linked together to have a functioning DARTHs solution. Deployments like those initiated with eWaterCycle or Deltares-FEWS (Werner et al. 2013) and LIS (Peters-Lidard, 2007) are examples of integrators of resources, maybe not fully compliant with the DARTHs architecture, that can be a good starting point to provide DARTHs and and an example of DARTHs providers.*

*Further ideas are expressed in Nativi et al. (2021) that has a specific section entitled: "Effective Governance of the Independent Enterprise Systems of a Digital Ecosystem" (in very technical language though).*

- C3.2 - How can a decision process in the (further) development of DARTHs be designed? Who decides what?

*A 3.2 - DARTHs are more a process than a tool needing a final stage of development and, as such, they are not per se subjected to "further development", intended as a separate step of their existence. DARTHs grow and develop continuously by adapting  to the emerging scientific questions. In principle there is no need for a centralized director for most of the developments. However, organized communities can clearly obtain more achievements. Any community, in particular, can choose its own policies that  will be  effective if it chooses to adopt common standards for data and formats, while also promoting innovative solutions itself. The Jules project ([https://jules.jchmr.org/](https://jules.jchmr.org/)) is an example of governance for such communities. Examples of successful  communities can be seen also in Archfield et al., (2015).  A bigger model to follow, outside the Hydrology,  is provided by the GNU/Linux project where one community is responsible for the kernel of the operating system and others, including GNU, provide additional tools and at the end of the process other subjects deploy variously flavored versions of GNU/Linux, called Linux distributions.*

- C 3.3 - Can/should DARTHs be non-commercial or does it need to be designed as a commercial endeavor?

*A 3.3 - The DARTH does not need to be either commercial or not commercial. Both strategies can be pursued (and, we guess, will be pursued). Please observe also that commercial is not the opposite of Open Source. An open source product can be commercialized and a commercial product can be open source.  So DARTHs can be both commercial and not commercial but require appropriate  apeness characteristics.*

- C 3.4 Or in other words: who can afford it, how can financing look like?

*A 3.4 - Building pieces of a DARTHs (a DARTH component), according to our ideas, is feasible by an organized group of researchers (especially at the Linker level), or partnerships funded by international research programs.  The OpenEO initiative, https://openeo.org, in the field of Earth observation data development is an example of how a DARTH development could be funded. The entire scope of this paper is to suggest a modular architecture whose science parts can, in principle, be affordable to many, if not all scientists, in the very same way that scientists gathers around organized modeling efforts as it was for JULES (Best et al., 2011), CLM (Lawrence et al., 2019), and other projects. SUMMA (Clark at al, 2015) and GEOframe (Formetta et al., 2014) are efforts of a single research group that can provide a starting platform for a DARTH. Certainly producing a DARTH from scratch would be a major effort that only large company associations or governments could afford.*

- C 3.5 - Who is responsible for quality control of items added by users?

*A 3.5 - In the end the responsible party is the community that builds a specific part of the DARTHs and the community or the commercial entity that Links the various DARTHS components to form a DARTH distribution. On the other hand, new techniques like the block-chain can be used to certify the steps and the chain of responsibility that a certain DARTH distribution has followed.*

*DARTHs initiatives shall also come up with test cases with specific links to (or support from) important experimental activities, sites and observatories (e.g., https://www.hymex.fr/liaise/index.html) that link experimental measurements and observations to model development. The same observatories can also be responsible to build shared benchmarking datasets to test DARTH performance.*

*If we refer to studies pursued with DARTHs, the ultimate responsibility is of the Runners that perform the studies, and eventually, the journal that publishes them. One critical aspect though, will be the availability of data, simulations records and software for allowing tests of results by other researchers.*

C 3.6 Who is responsible for avoiding misuse and misinterpretation of the models/data by non-specialists (e.g players, viewers and to some extent perhaps also runners)?

*A 3.6 - Whoever misused. But actually this question implies that there is a subsequent action with respect to the execution of the DARTH, which is the publication in some form of the results. In this case, whoever publishes is responsible. Moreover, experience from the pandemic of SARS-COVID2 suggests that the reader must be educated and that keeping information hidden does not solve issues and does not cultivate the credibility of science. On the other hand, the presence of the errors estimation, the replicability of the experiment and the open code are landmarks for the good will of acting fairly.*

*The outcomes of this discussion will be added to the final version of the manuscript or in its supplemental material.*

**C 4** - To grasp the context of DARTHs and its evident differences to other previous and ongoing initiatives that are currently state-of-the-art in our discipline (e.g. modular modeling frameworks), it will be helpful for the reader if authors provided a bit more detail in the discussion of similarities/differences with at least a few very *specific* other modeling frameworks. This could include comparisons with modular modeling frameworks at various levels of complexity such as SUPERFLEX (e.g. Fenicia et al., 2011) or SUMMA (e.g. Clark et al., 2015) and further extend to more versatile tools such as the very recently introduced eWaterCycle platform (Hut et al., 2021) – which is, in my understanding already quite a large step towards DARTHs. I believe a very simple table in which it is indicated which of the

currently available tools already tick which boxes and which additional boxes DARTHs could tick.

*A 4 - It is very difficult to give judgment on the work of others and the operation is prone to give very biased responses. To our knowledge, many projects have made substantial efforts towards the DARTHs requirements, but DARTHs still require further software architectural efforts that cannot come only from the hydrological community. With respect to how much the requirements are fulfilled by current platforms, we have prepared a questionnaire and submitted to the colleagues who implemented the cited systems and others,and whose results highlight the closeness of their systems to the DARTHs ideals. The questionnaire is available at https://form.jotform.com/220481340392348, and will be made available through the blog AboutHydrology and as supplemental material to this paper. The results, present at https://eu.jotform.com/tables/220481340392348 will be collected and presented, if available before the review processes, as supplemental material too. In any case we will try to make some specific examples directly in the manuscript and to rewrite more clearly some critical parts (as the definition on MaaA, MaaT, etx.)*

**C- 5.** The language becomes a bit too informal in parts of the manuscript and could benefit from being more precise to avoid ambiguities.

*A- 5- We will revise the language trying to eliminate colloquial phrasing.*

**Detailed comments:**

p.2, l.30: I suggest replacing "understanding" with "describing", as "understanding" is part of the discovery process and thereby meta-science. Whether or not you personally understand something is not really relevant (and there is of course no "collective" understanding). At one point something clicks in your brain but how is that relevant for other people? In other words, it remains something very subjective (and thus the opposite of what science should be).

*A - It is true that the understanding has some meta-scientific meaning. But it is exactly what we meant.*

p.2, l.39: Should probably read "Space Agencies" instead of "Spatial Agencies"

*A - It will be corrected in the new version of the work.*

p.2, l.42ff: the use of "top-down" and "bottom-up" may generate confusion as they are typically used for very specific modeling strategies in hydrology/environmental sciences. Perhaps helpful to use a slightly different terminology here.

*A - We rephrased as: " where research and its related data are created and shaped by big institutional players, even private entities' '. We could rephrase it to "where research and its*

*related data are created and shaped by big institutional players, even private entities, from the "top" down to the researchers"*

p.2, l.43ff: I found this a statement that is a bit too sweeping, generalizing and pessimistic. There are many research groups that actively work on model development/improvement. And any other research group that does not, is of course free to start working on this anytime. It reads as if these poor people are forced to use models imposed onto them by some higher force. Perhaps good to tone it down a bit.

*A - The sentence will be rephrased in the following way: "As such, this top-down approach could limit creativity and the possibility of a vast community to contribute to the advancement of science and innovation (Oleson et al., 2013; Best et al., 2011)."*

p.3, l.61: do you really intend to already "answer" these questions? This seems a bit too ambitious and restrictive. I think can be reformulated into something like "outline potential ways forward"

*A - In Accordance with Dr. Hrachowitz,, we will change as suggested: "debate and outline potential ways forward:" or "debate and try to answer the following questions:"*

p.3, l.65: the term "certified" seems a bit awkward here. Not clear what you mean to say here.

*A - The term certified here is somehow related to the Reviewer's first comment. The "certification" should mark good science, which we are presenting and debating in this work, with respect to bad science.*

p.3, l.72: what is meant by "reasonable color maps"? Beautiful maps or maps that show plausible patterns? Similarly, what is meant by "…have no…issues"? how do you define issues? Depending on the definition, one could equally say, that *all* models have a lot of issues. Please rephrase.

*A - Here we meant maps that show plausible patterns, obtained by using models that are solid and reliable from the scientific point of view, broadly used and tested, which present few or no problems from the end-user's point of view. This sentence will be rephrased as " plausible patterns".*

p.3, l.74: I strongly disagree. That is what for example the many recent modular frameworks are aimed at.

*A - We will add to the paragraph as follows: " but they are often closed to easy modification and lag behind the state of the art of hydrological studies. The state of art …. require. Many modular frameworks have been recently built with the aim tof filling this gap. However, from … to cope with the DARTHs requirements. "*

p.3, l.79ff: Meaningful classification of models is indeed tricky but I believe the taxonomy provided here does not really capture the main differences in model features. The main differences between models, as we argued in Hrachowitz and Clark (2017), are the level to which physical constraints are imposed. For example, typically data-driven/statistical/machine-learning models (notwithstanding some recent developments) have not even imposed conservation of mass. Conceptual/reservoir –type models at least satisfy that constraint and work with a few additional process assumptions. The level of process representation then increases towards models, such as ParFlow which of course have much more detailed process representations. Therefore I would rather refer to all models that use at least some process assumptions as process-based on a gradual spectrum. In addition, I believe referring to lumped model implementations here can also spark some confusion. No matter which model type is used – it can be implemented at any spatial resolution. If this is justifiable is of course a completely different question. Perhaps try to reformulate this paragraph.

*A - What Dr. Hrachowitz says it is arguably true, however, we were not trying to classify models on the basis of "being more or less physically based" but rather on the type of mathematics they use (partial differential equations, ordinary differential equations, traditional statistics, machine learning). The reference to black-box models was also trying to capture the EO products that contain internal modeling: as this part is better explained in the specific section, we can omit from here. We will revise the paragraph as follows:*

*"Broadly speaking, four **mathematical tools** dominate in hydrological modeling research: modeling by partial differential equations (often called, process-based, PB) (Paniconi and Putti, 2015; Fatichi et al., 2016a); the reservoir type models (HDS, as in Hydrological dynamical systems) (Todini, 2007; Bancheri et al., 2019b); the classical statistical models (McCuen, 2016); and the current algorithmic statistical models that make use of some form of machine learning (Shen et al., 2018; Levia et al., 2020). To these we could add a further type of model which is a tight, black-box, called EO products that merge together EO and some type of hydrological modeling (Martens et al., 2016). **Many references already discuss the taxonomy of models (e.g. Kampf and Burges, 2007)**, the strengths and weaknesses of each of the approaches (**Hrachowitz and Clark, 2017**) and their application at various scales, and we do not want to add further. The paper in fact aims to clarify more some software architectural needs and some workflow practices than establish models contents aspects."*

p.4, l.86: not clear what is meant here.

*A - Here we mean that the cited models are used broadly by a vast community of hydrologists for many interesting applications, however, we are not sure about how robust the implementations are from the informatics point of view.*

p.4, l.89: what is meant by "panorama"?

*A - We revised to "models variety"*

p.4, l.90: System complexity emerges to quite some degree from variability and heterogeneity. They are therefore intimately linked. However, the way it is expressed here gives the impression that "complex" and "complicated" constitute some sort of dichotomy, as in "on the one hand and on the other hand", while it should rather be that one follows from the other.

*A - "complex" and "complicated" as used in l.90 are not intended as a dichotomy but neither are they one the consequence of the other. They are actually two distinct concepts: complicated, here, means not easy and, as we stated, variable and heterogeneous. However, "complicated" doesn't mean that it could not be solved. Complexity means tangled, based on multiple processes not always easy to be derived, separated, analyzed individually and solved.*

p.4, l.100: I do not really understand that statement. Of course ML can be "investigated". Why should this not be possible? It is a human-made construct. As such it can be adjusted but also looked into. I guess you mean until recently it was difficult for non-experts to analyze what is happening in the code of ML models. Please make this clearer. In addition, I do not believe that ML can have "knowledge". Makes it sound like a conscious entity, which it is (to my knowledge) not yet. Please rephrase.

*A - Usually ML and especially Neural Networks and Deep learning are seen as "black-box" tools whose internals are impossible to disentangle. The paper we have cited illustrates some techniques to do it. We will change the phrase from:*

*Recently, in fact, it has become clear that ML can be investigated and its knowledge explained. It has also become clear that Deep Learning (DL) techniques can be used as tools of interpretation (Arrieta et al., 2020) or model parameters learning (Tsai et al., 2021) instead of primarily for predictive purposes.*

*to :*

*Recently (e.g. Gharari et al., 2021), in fact, it has become clear that ML and Deep Learning (DL) techniques can be interpreted (Molnar et al., 2018, 2019) and explained thus they can be. used as a tool of interpretation (Arrieta et al., 2020) or model parameter learning (Tsai et al., 2021) instead of primarily for predictive purposes.*

p.5, l.134: this is not unique for the US. Environmental data from many European countries are also publicly and readily available. For example, Austria (e.g. https://ehyd.gv.at/), France (e.g. https://www.hydro.eaufrance.fr/), Germany (e.g. www.dwd.de), UK (e.g. https://archive.ceda.ac.uk/), and many others.

*A - We will add to the text: " ... as well Austria (e.g. https://ehyd.gv.at/), France (e.g. https://www.hydro.eaufrance.fr/), Germany (e.g. www.dwd.de), UK (e.g. https://archive.ceda.ac.uk/), and many others."*

p.6, l.166: not clear what is meant by "binding".

*A -Here binding means collating data from different sources and providers. The word "binding" was added to the Glossary.*

p.6, l.178 (and elsewhere): please clarify the meaning of "seamless" here

*A- smoothly*

p.8, l.212ff: I found this paragraph very difficult to follow and I am not sure what the authors try to express here. Perhaps helpful to reduce jargon or to explain in a bit more detail.

*A - To facilitate the understanding of the paragraph, we will rephrase it all and add a glossary with the terms. Of course, the difference between necessary technical language and jargon is sometimes subtle and in the revised text we will try to stay on the side of the best possible understanding. We will rephrase the lines between 210 and 220 as follows:*

*"In order to include their own software in a DARTH infrastructure, developers must rely on some APIs (Application Programming Interfaces), i.e. software intermediaries that allow communication between the developers' software and the DARTH infrastructure. Therefore, a DARTH infrastructure that serves the needs of everyone should be able to accept different modeling styles and paradigms, changing them only if required and, in general, not being invasive of programming habits (i.e. not forcing the programming towards constructs that computer scientists like but scientists and engineers cannot manage). However, the legacy within the infrastructure can be a potential critical point (bottleneck) in the development of a DARTH model. The available environmental modeling frameworks, infrastructures to all effects, can be classified into two broad categories: heavyweight frameworks and lightweight frameworks (Lloyd et al., 2011). The former are characterized by large and unwieldy APIs that require considerable effort for developers (scientists or soft coders) to become familiar with before writing new code. This effort becomes even more demanding if one considers that research groups often maintain many legacy models based on algorithms and equations developed decades ago (David et al., 2014). Moreover, such an effort somehow creates a strict legacy within the infrastructure, and this limits the possibility of having more than one modeling solution in the same DARTH."*

p.9, l.261: perhaps replace "reality" with "real world observations". In addition, please specify what is meant by "internals".

*A - In the next version of the work, the Authors will change reality with "real world observations". A model's internals refers to the implemented equations constituting the model itself.*

p.9, l.262: but this needs to be a very detailed knowledge of the simulation set-ups as demonstrated by Ceola et al. (2015) and generally argued to be impossible by others (e.g. Hutton et al., 2016). Please tone down and reformulate.

*A - Ceola et al.addresses different issues to the one we refer to here. Ceola's paper argues about replicating results with the same model but with two different cases for parameter setting: a first using a random procedure for assessing a model's parameter, and a second case letting people vary parameters and the workflow according to their experience. Ceola's paper, as such, deals with the common workflow in modeling. (Obviously, when Ceola et al. workflow includes stochastic searches of parameters, they cannot replicate their finding twice by definition)*

*Our effort instead is in the direction of those aspects partially highlighted by Knoben et al., 2021, since the issue we deal with in our paper is related to the software architectural implementation of characteristics that simulations need in order to be replicable. The bottom line of our arguments is that for reproducibility of a certain workflow, it is necessary to record it. Ceola et al. have a protocol of actions to be fulfilled to run properly, but adherence to the protocol becomes a matter of common agreement between fair runners as no record of compliance remains.*

*We will modify the text to make these concepts more clear.*

p.10, l.274: what is meant by "building tools"?

*A - It is actually build tools, which are programs that automate the creation of executable applications from source code. Building incorporates compiling, linking and packaging the code into a usable or executable form. Please see:* https://en.wikipedia.org/wiki/Software_build. *We will add the exaplanation of what "building tools are in the Glossary.*

p.10, l.275: what does "…to certify the providers of models…" mean and entail?

*A - Here certify means to validate the goodness of the providers of the models, such as through publications.* *We will rephrase the sentence, clarifying the concept.*

p.10, l.283: please specify "all they need"

*A - We will do it, they are input data, parameters values, output file names, modeling solution structure. See also the answer below.*

p.10, l.285: what does the "prepared simulation" include? Calibration set-up? Results? In addition, what is meant by "governed"?

*A- The already prepared simulation is described and governed by ".sim" files in the "simulation" folder, which contains the model components used, their connections in a workflow, their parameters' values, indications of which files to read as input and which output to produce (e.g. David et al., 2009).* *We will specify it in the revised manuscript.*

p.10 or elsewhere: I am not sure where this fits in, but one aspect that seems crucial to me is the definition of the smallest, unchangeable building block of models in the entire system.

What could these be? Can there be multiple? Who decides on that? Can users (e.g. Runners) just add such building blocks and/or specific parameterizations (as in reality we have no idea which parameterizations – i.e. equations, not parameter values! – are most suitable where/under which condition/at which scale/etc. see e.g. recent analysis by Gharari et al., 2021)

*A- According to our definition of DARTHs, the smallest build block is a component. Components are self-contained modules or units of code. Each well-designed "component" usually implements a single modeling concept. Components can be joined together to obtain a modeling solution that can accomplish a complicated task, such as simulating the water budget storages and fluxes of a catchment. Multiple algorithms can be implemented within the same component or in various components, and inserted into modeling solutions as alternatives, thus opening the way to compare different approaches within the same chain of tools. We will add this information in the glossary. Clearly the concept can also be implemented by modularizing the code of a model but components add much more flexibility: for instance, they can be added or substituted to equivalent others (in term of inputs and outputs) without opening the original source code and recompiling it. They can be authored by different programmers from the ones producing most of the code and everyone thus can see their contribution better recognized. They encapsulate their contents much better than possible within a class or a subroutine and they can be run independently and alone, provided the appropriate inputs, for testing and quality control. They can be maintained independently of the rest of the code, and so on.*

p.11, section 5: although I have a fair share of model development/coding experience, I struggled with the entire section. Frankly, I could not follow it. In particular, it was difficult to grasp what the subtle (or perhaps for the specialist not so subtle) differences between the five classes MaaA, MaaT, MaaS, MaaR and MaaC are and what follows from these differences. For example it would be very instructive and helpful if you could let the reader know into which class different existing models, modular frameworks and platforms fall (e.g. SUPERFLEX, HYPE, SWAT, SUMMA, eWaterCycle)

*We believe that the description of other Authors' code is always biased. To make this more objective, we have prepared a questionnaire with a double purpose. The first is that by looking at the questions, the readers can better understand what the classification of models provided by the acronyms above means and, second, should they answer the questions, they can understand where their own model or a specific model they know is placed. The survey can be found here: https://form.jotform.com/220481340392348 and the answers are here: https://eu.jotform.com/tables/220481340392348.* Survey results and comments will be provided as supplementary material to the paper. The description of MaaA, MaaT, MaaS, MaaR, MaaC and their differences will be made clearer.

p.12, l.326: The role of the provider remains quite vague. Is this the data provider? Is this the developer of the model concept/idea? Is it the developer of a model code that is based on a specific model concept/idea? Is this somebody completely different? Also the term "policies" is unclear here.

*A- A provider is an entity that provides the data necessary to run the models. It will be rephrased as: " Within MaaT everything is controlled by the developer and the funding subject of the data, who not only establishes the policies for the use of models, in terms of its sharing and usage, but also controls model evolution and enhancement. "*

p.14, l.365: "invasiveness"??

*A- According to Lloyd et al., 2011, "non-invasive" means that, with respect to other similar frameworks, OMS does not change the habits of a good programmer. In particular, to be used as an OMS3 component, a Java class needs only to be enriched with annotations. Other frameworks, for instance OpenMI (Gregersen et al., 2007), require a special style of programming which modifies what programmers usually do. Just to cite an example, the same task written in Java for OPENMI 1.2 can contain 1.5 times the code than that for OMS3. For a clear understanding of the concept, please refer to Lloyd's paper.*

p.14, l.380: "…not confined to convey science from a single discipline." Sounds awkward. Please rephrase

*A- It will be rephrased as "and components can accomplish a wide range of tasks, not necessarily from the same discipline"*

p.14, l.384: what are "fake" models? Models that do not exist? Please rephrase.

*A- "Fake models" refers to models that are not robust from the scientific point of view. In this sense, the reliability of the codes must always be proved, carefully tested against real data and their uncertainty should be analyzed. It will be rephrased as "since it can potentially lead to the spreading of unreliable or untested models "*

*p.14, l.388: "under the hood". Please rephrase.*

*A - It means models' internals. It will be rephrased as "The DARTH metaphor requires an extremely large use of data exchange in the background, which requires extremely high computational power."*

p.15, l.462: "some conditions" is quite an understatement. With our currently available observation technology *most* process dynamics and system properties (e.g. soil hydraulic conductivities) are unknown at most locations during most of the time – in reality we have no idea of the spatial covariance fields of most of these quantities. Instead and to deal with this

problem we make sweeping assumptions about this missing information and thereby we very likely upscale homogeneity instead of heterogeneity.

*Certainly true. We took away "some": Part of the uncertainty certainly comes from ignorance of conditions that, being unknown, yet necessary to completely define the mathematical problem, must be guessed.*

*p.15, l.464: well, not only data errors, also model structural errors can and do result in parameters that do not reflect real world system properties.*

*A - In fact, it is written: "Errors in measurements affect the procedure of calibration and cause the inference of incorrect model parameters. Errors in model structure reflect in wrong forecasts, which in turn cause biased comparisons between them and the models outputs." The question of parameters remained quite implicit though and we will try to write it better.*

p.19, l.480: is the range of results really that restricted? How is it then possible that different models exhibit considerably different (internal) behaviors (e.g. Bouaziz et al., 2021)?

*A - We have removed "quite".*

p.19, l.480ff: "some type of warning": this is extremely relevant and deserves some more consideration and detail in the text.

*A- Is not what we wrote below enough: "Among the myriad procedures for calibration, sensitivity analysis, and data assimilation, error estimation is more art than science: while the methods are rigorous, the assumptions under which they work are of varying credibility depending on the process. For instance, Refsgaard et al. (2007), recognizes at least 14 methodologies (plus GLUE) to obtain these estimates. Here, we do not advise the use of any particular one of these methods, but we do claim that at least one method should be chosen. Moreover, we want to reinforce the idea that error estimation is a practice that has to be continuously exerted and refined. If the comparison between computed and measured quantities is systematically done in DARTHs, then statistics of the performance of a certain model setup becomes more reliable with time." ?*

*Anyway, we will change "some type of warning" to "appropriate warnings". Besides we modified also the phrase below l. 483 to make more precise the concept.*

p.19, l.483ff: perhaps also good to refer to the exchange between Nearing et al. (2016) and Beven (2016), which is very reflective of these issues that are yet unsettled.

*Thanks for the suggestions. We will add the citations of the two papers. Besides, in the subsequent phrase we will add a phrase almost literally taken from Beven (2016).*

*We will modify the text as follows:*

*Here, we do not advise the use of any particular one of these methods, but we do claim that at least one method should be chosen.* **If, for the sake of science advancement, the search for the origin of errors is paramount, for what regards DARTHs we stay with the simplest fact that "purely empirically, probability and statistics can, of course, describe anything from observations to model residuals, regardless of the actual source of uncertainty (Cox, 1946)".**

p.21, l.529-548: very interesting and important ambition!!

*A - Yes, sure !*

p.22, l.555: as recently also demonstrated by e.g. Gharari et al. (2021): given the limited observations we have relative to the size and complexity of our systems, process-based (i.e. "conceptual" and "physically-based") models can too restrictive with their assumptions on the type/shape of functional relationships.

*A - We thank the reviewer for pointing us to the paper by Gharari which had escaped our attention. We will add the reference as a valuable point of view on model complexity and modify the text appropriately.*

Above I have added quite a few references of our group. Please see them as mere examples and suggestions. It was only done for convenience (easier for me to find our references than those of other groups). Needless to say that many other groups work on similar topics and their references may be more suitable. Therefore, please do not feel obliged to use the references suggested here.

*A - Thank you they were very valuable.*

Best regards,

Markus Hrachowitz

References by the Reviewer:

Beven, K. (2016). Facets of uncertainty: epistemic uncertainty, non-stationarity, likelihood, hypothesis testing, and communication. Hydrological Sciences Journal, 61(9), 1652-1665.

Bouaziz, L. J., Fenicia, F., Thirel, G., de Boer-Euser, T., Buitink, J., Brauer, C. C., (2021). Behind the scenes of streamflow model performance. Hydrology and Earth System Sciences, 25(2), 1069-1095.

Ceola, S., Arheimer, B., Baratti, E., Blöschl, G., Capell, R., Castellarin, A., ... & Wagener, T. (2015). Virtual laboratories: new opportunities for collaborative water science. Hydrology and Earth System Sciences, 19(4), 2101-2117.

Clark, M. P., Nijssen, B., Lundquist, J. D., Kavetski, D., Rupp, D. E., Woods, R. A., ... & Rasmussen, R. M. (2015). A unified approach for processâbased hydrologic modeling: 1. Modeling concept. Water Resources Research, 51(4), 2498-2514.

Fenicia, F., Kavetski, D., & Savenije, H. H. (2011). Elements of a flexible approach for conceptual hydrological modeling: 1. Motivation and theoretical development. Water Resources Research, 47(11).

Gharari, S., Gupta, H. V., Clark, M. P., … & Savenije, H. H. (2021). Understanding the information content in the hierarchy of model development decisions: Learning from data. Water Resources Research, 57(6), e2020WR027948.Hrachowitz, M., & Clark, M. P. (2017). HESS Opinions: The complementary merits of competing modelling philosophies in hydrology. Hydrology and Earth System Sciences, 21(8), 3953-3973.

Hut, R., Drost, N., van de Giesen, N., van Werkhoven, B., Abdollahi, B., Aerts, J., ... & Weel, B. (2021). The eWaterCycle platform for Open and FAIR Hydrological collaboration. Geoscientific Model Development Discussions, 1-31.

Hutton, C., Wagener, T., Freer, J., Han, D., Duffy, C., & Arheimer, B. (2016). Most computational hydrology is not reproducible, so is it really science?. Water Resources Research, 52(10), 7548-7555.

Nearing, G. S., Tian, Y., Gupta, H. V., Clark, M. P., Harrison, K. W., & Weijs, S. V. (2016). A philosophical basis for hydrological uncertainty. Hydrological Sciences Journal, 61(9), 1666-1678.

*Weiler, M., & Beven, K. (2015). Do we need a community hydrological model?. Water Resources Research, 51(9), 7777-7784.*

**References by Authors**

Archfield, Stacey A., Martyn Clark, Berit Arheimer, Lauren E. Hay, Hilary McMillan, Julie E. Kiang, Jan Seibert, et al. 2015. "Accelerating Advances in Continental Domain Hydrologic Modeling." *Water Resources Research* 51 (12): 10078–91.

*Best, M. J., M. Pryor, D. B. Clark, G. G. Rooney, R. L. H. Essery, C. B. Ménard, J. M. Edwards, et al. 2011. "The Joint UK Land Environment Simulator (JULES), Model Description – Part 1: Energy and Water Fluxes." Geoscientific Model Development 4 (3): 677–99.*

*Clark, Martyn P., Bart Nijssen, Jessica D. Lundquist, Dmitri Kavetski, David E. Rupp, Ross A. Woods, Jim E. Freer, et al. 2015. "A Unified Approach for Process‑based Hydrologic Modeling: 1. Modeling Concept." Water Resources Research 51 (4): 2498–2514.*

*David, Olaf, Jack R. Carlson, Lajpat R. Ahuja, Frank W. Geter, and George H. Leavesley. 2009. "Object Modeling System V3.0 - Developer and User Handbook," August, 1–103.*

*Formetta, G., A. Antonello, S. Franceschi, O. David, and R. Rigon. 2014. "Hydrological Modelling with Components: A GIS-Based Open-Source Framework." Environmental Modelling & Software 55 (May): 190–200.*

*Gregersen, J. B., P. J. A. Gijsbers, and S. J. P. Westen. 2007. "OpenMI: Open Modelling Interface." Journal of Hydroinformatics 9 (3): 175–91.*

*Knoben, Wouter Johannes Maria, Martyn P. Clark, Jerad Bales, Andrew Bennett, S. Gharari, Christopher B. Marsh, Bart Nijssen, et al. 2021. "Community Workflows to Advance Reproducibility in Hydrologic Modeling: Separating Model-Agnostic and Model-Specific Configuration Steps in Applications of Large-Domain Hydrologic Models." Earth and Space Science Open Archive. https://doi.org/10.1002/essoar.10509195.1.*

*Kumar, S.V., C.D. Peters-Lidard, Y. Tian, P.R. Houser, J. Geiger, S. Olden, L. Lighty, J.L. Eastman, B. Doty, P. Dirmeyer, J. Adams, K. Mitchell, E. F. Wood, and J. Sheffield, 2006: Land Information System - An interoperable framework for high resolution land surface modeling. Environ. Modeling & Software, 21, 1402-1415, doi:10.1016/j.envsoft.2005.07.004*

*Lawrence, David M., Rosie A. Fisher, Charles D. Koven, Keith W. Oleson, Sean C. Swenson, Gordon Bonan, Nathan Collier, et al. 2019. "The Community Land Model Version 5: Description of New Features, Benchmarking, and Impact of Forcing Uncertainty." Journal of Advances in Modeling Earth Systems 11 (12): 4245–87.*

*Peters-Lidard, C.D., P.R. Houser, Y. Tian, S.V. Kumar, J. Geiger, S. Olden, L. Lighty, B. Doty, P. Dirmeyer, J. Adams, K. Mitchell, E.F. Wood, and J. Sheffield, 2007: High-performance*

*Earth system modeling with NASA/GSFC's Land Information System. Innovations in Systems and Software Engineering, 3(3), 157-165, doi:10.1007/s11334-007-0028-x*

*Werner, M., J. Schellekens, P. Gijsbers, M. van Dijk, O. van den Akker, and K. Heynert. 2013. "The Delft-FEWS Flow Forecasting System." Environmental Modelling and Software[R] 40 (February): 65–77.*

---

## Author Comment (AC2)

**Answer to Comments to the Authors by Reviewers of "HESS Opinions: Participatory Digital Earth Twin Hydrology systems (DARTHs) for everyone: a blueprint for hydrologists' by R. Rigon et al.**

**Comments by Dr. Uwe Ehret**

**C 1**- Dear Editor, dear Authors, I have reviewed the aforementioned work. My conclusions and comments are as follows:
1. Scope
The article is well within the scope of HESS.

2. Summary
In their opinion article, the authors describe both the necessity and the key aspects of moving hydrological modeling from their current state towards Participatory Digital Earth Twin Hydrology Systems (DARTHs). Key aspects of such systems include i) flexible coupling to many sources of data available in standardized formats , ii) platform- and language-independent implementation, iii) modular design facilitating recombination of model components. Each such aspect is explained in detail in a separate section and briefly summarized in the conclusions.

3. Evaluation
Overall, this article is a relevant and timely contribution to the discussion about the future of earth system modeling in general and hydrological modeling in particular. It discusses many current hindrances towards efficient, integrated and collaborative modeling and names key elements of a framework to overcome these hindrances. So I welcome the article for its content,

*A -1 - We thank the reviewer for the appreciation of our paper and we will do our best to accommodate his valuable suggestions.*

**C 2** but I do have several suggestions to improve its presentation:

Key messages: For a reader not strongly familiar with the current state of development of Digital Earth (DE) and Digital Twin Earth Models (DT), only after reading the entire manuscript it becomes clear what the differences to current hydrological modeling practice are, and consequently what the authors aim at with their article. A reader will be able to follow the arguments in the paper much better if this is made clear from the beginning. I therefore suggest the following changes to the manuscript:
- Include in the introduction a clear definition (purpose, key structural and organizational elements) of DE's, DT's, DARTH's and current hydrological models. Make clear where the latter fall short of qualifying as DARTHs, why it is necessary to elevate them towards becoming DARTHs, and what it takes to do so. A comparative list, or an explanatory chart can support this (maybe directly pointing to the sections

in which the individual topics will be discussed in detail). From such an introduction which provides the overall picture, it will be easier for the reader to connect to the individual sections.

*A 2 - We will try our best to consider the Reviewer's suggestions in revising the introduction and to provide a better overview in the Abstract. However, we must also be careful not to overwhelm the reader at the outset with an overload of information in the Introduction. . Certainly, we will provide the required definitions and a short overview of the subsequent sections in the Introduction, as well as a Glossary explaining the main technical terms appearing in the paper.*

**C3 -** At the beginning of each topical section (sections 2-9), include a very short overview of the section content.

*A 3- We will probably do this at the end of the Introduction*

**C 4** - Abstract: From reading the abstract only, I did not have an idea about what to expect from the article, and what its key messages are. For example, I did not know how to interpret the term "model as commodity", which is not standard terminology in hydrology. Please rewrite, taking into consideration my previous comment about key messages.

*A 4 - Yes, of course. We will modify the Abstract to give a clearer overview of the paper.*

**C 5** *- L271: I partly agree with the statement "The peer review process is ineffective at sniffing out poor model validation", but while it may be far from perfect, I'd say it is still the best we have. Please explain how in a DE/DT/DARTH world quality standards are maintained better than by peer control.*

*A 5 - The statement was certainly not aimed against the peer review process itself. We recognize, however, that even when code is Open Source it is difficult to find mistakes in models. In fact, we would go so far as to say that it is impossible without: a) proper software organization; b) simulations setup tracking and recording; c) clean code writing; and d) sufficient further documentation. Open Source code, however, can be inspected and errors potentially caught by third parties. The above characteristics in principle are not required in a DE/DT but we require them in DARTHs to improve the science check and reproducibility.
We will modify the text by adding "alone": "The peer review process **alone** is ineffective at sniffing out poor model validation"*

**C 6** Section 7: The section header just mentions "reliability", but in the text more aspects of model performance are discussed. I suggest widening the section header, and discussing robustness, reliability, reproducibility and realism (as mentioned in L260).

*A 6- We think that enlarging the scope of the section beyond the aspects already analyzed would require another paper. Reproducibility is discussed throughout the paper and realism*

*of the model is certainly related to what was discussed in the section . We might consider changing the title of the section. However, we will give a clearer definition of "robustness, reliability, reproducibility and realism" in the revised text and we will indicate where we have discussed the topics.*

**C 7** - Section 10: The conclusions are written in a rather hasty and bullet-point style, which devaluates the otherwise interesting and comprehensive paper. Please spend some more effort in writing a coherent and standalone summary and conclusion.

*A 7 - Conclusions will be revised in the next version of the work.*

**C 8** - Appendix A: Reading Appendix A, it was not clear to me why this section was moved to the appendix rather than being another section in the main text. I'd say the topic of proper coding is just as relevant for developing DARTHs as the other topics in sections 2-9. I suggest moving it to the main text.

*A 8 - It would be easy to move this Appendix back to the main text. However, its character, in our opinion, is a little different from the other sections, and we prefer to keep it where it is.*

*Minor points*

- L72: What is meant by "reasonable colour maps"?

*Here we meant maps that show plausible patterns, obtained by using models that are solid and reliable from the scientific perspective, broadly used and tested, and that present few or no problems from the end-user's point of view. This sentence will be rephrased as " plausible patterns".*

- L82-84: It is unclear at this point what is meant by "tight, black-box models", or rather how they differ from the previously mentioned models. Please clarify.

*Black-box models are those models in which the internals are hidden in a "black-box", and the only parts that can be viewed are the inputs and outputs.*

- L89-90: The meaning of the sentence ("At present, …") is not clear to me. What is meant by "modeling panorama" here? Please clarify.

*In this sentence we mean that there are a lot of models available with different levels of complexity, making it difficult to simplify the matter. We changed "modeling panorama" into "models variety"*

- L203: For me, the user's names are unfortunately not self-explanatory. Please add a short explanation similar to the explanation of roles.

*Users are actually the runners, players, viewers and providers. The sentence is misleading and it will be dropped.*

- L353: Please explain the "information hiding principle", because it seems to contradict the general paradigm of openness and accessibility advocated throughout the paper.

*Here "information hiding principle" refers to the encapsulation principle, which in object-oriented programming prevents direct access to objects by clients to avoid exposing hidden implementation details or violate state invariance, maintained by the methods. This principle is not preventing the openness and accessibility, since it is a runtime process. However, we will add a glossary to define all these concepts.*

- Caption of Fig. 5: What is meant by "IT"? Also, please add the channel network to the upper map for better connection to the lower map.

*"IT" is a typo: it should read "It". The network will be added to the upper plot.*

- *Table B1 is very helpful. Please mention it already in the introduction, such that the reader can make use of it while reading the paper.*

*It will be done in the next version of the work*

- Overall, there are many typos in the manuscript. Please check and remove.

*A careful editing will be done on the work.*

**C 9** -I have another question to the authors related to current hydrological modeling and hydrological modeling in a DARTH environment. There is no need to address this in the manuscript, but I am interested in the authors's opinion. The question is related to emergence as larger-scale phenomena arising from strong interactions on smaller scales. Current models build for a specific purpose and a specific scale/resolution often either make implicit use of emergence (by taking emergence as a given, directly representing the emerged phenomenon) or, because it may be relevant, setting the model up in a way that emergence may actually happen as the model runs (e.g. emergence of convective thunderstorm cells in convection-permitting atmospheric models). The latter often requires a well-attuned choice of time-stepping, spatial resolution, numerical scheme, variable precision, processes represented, etc. With a free and modular combination of model components in a DARTH environment, we may miss such well-attuned combinations, which may lead to not only

incrementally but fundamentally different model behavior. How can this be taken care of in DARTH environments?

*A 9 - The question raised by Dr. Ehret is important. In DARTHs constructions there are two aspects to be aware of. One is the infrastructure: the rules and the general principle of the organization of the infrastructure should be such that the infrastructure is agnostic to any science content. The other aspect is the science content.*

*In being agnostic, the DARTHs infrastructure is designed to accommodate as much as possible any science solution and allow its testing. The idea is to contrast the "hammer-nail" attitude (If the only tool you have is a hammer, you tend to see every problem as a nail.), present also in research, and to provide a very adaptable infrastructure in which any problem can find its proper solution (and not a solution that is shoehorned to the problem). The flexibility of infrastructure serves to test alternative hypotheses fairly and compare them against data evidence. The emphasis on tools establishing degrees of uncertainty is functional to selecting the more successful hypotheses, whichever they are, and eventually exclude those proving to be less reasonable.*

*Therefore DARTHs can evaluate any hypothesis, including the ones that involve changes in scales of the analysis.*

*Going to the science questions, we recognize that a statistical mechanics of hydrological phenomena is largely missing; however, the matter of emerging properties in hydrological models is the object of investigations that DARTHs could and should allow more easily than with traditional modeling. There is no problem in performing all the analyses that a sharp researcher would do with traditional tools. The Developers can implement new parameterizations more easily, the Linkers can explore the pool of model components, and the Users investigate the parameter space. The presence and use of tools to assess the error of estimation should eventually raise red flags when something is not properly modeled and warn the researcher to rethink the whole modeling process at their appearance.*

*One aspect that we probably did not stress enough is the fact that DARTHs impose a certain systematicity in the analyses (DE were said to be an "organizing metaphor"), the current absence of which is one of the obstacles to assessing laws of general validity. Recognizing emerging patterns and features should be easier when dealing with a multiplicity of catchments and from the many points of view that DARTHs promote.*

*DARTHs obviously do not substitute human creativity in formulating hypotheses, they only offer a wider range of tools to implement and test them appropriately.*

Yours sincerely,

Uwe Ehret